# Learning Robust Multi-Agent Policies via Selective Adversarial Fault Induction

**David Mguni** [1]  **Yaqi Sun** [1]  **Haojun Chen** [2]  **Wanrong Yang** [3]  **Amir Darabi** [4]  **Larry Olanrewaju Orimoloye** [4]
**Yaodong Yang** [2]

## Abstract

We study robustness to agent malfunctions in cooperative multi-agent reinforcement learning (MARL), a failure mode that is critical in practice yet underexplored in existing theory. We introduce MARTA, a plug-and-play robustness layer that augments standard MARL algorithms with a Switcher–Adversary mechanism which selectively induces malfunctions in performance-critical states. This formulation defines a fault-switching $(N + 2)$-player Markov game in which the Switcher chooses when and which agent fails, and the Adversary controls the resulting faulty behaviour via random or worst-case policies. We develop a Q-learning-type scheme and show that the associated Bellman operator is a contraction, yielding existence and uniqueness of the minimax value, convergence to a Markov perfect equilibrium. MARTA integrates seamlessly with MARL algorithms without architectural modification and consistently improves robustness across Traffic Junction (TJ), Level-Based Foraging (LBF), MPE SimpleTag, and SMAC (v2). In these domains, MARTA achieves large gains in final performance of up to **116.7%** in SMAC, **21.4%** in MPE SimpleTag, and **44.6%** in LBF, while significantly reducing failure rates under train–test mismatched fault regimes. These results establish MARTA as a theoretically grounded and practically deployable mechanism for fault-tolerant MARL.

## 1. Introduction

In multi-agent systems (MAS), agents must anticipate one another's actions and coordinate effectively in order to solve tasks (Albrecht et al., 2024). A fundamental assumption underlying most successful multi-agent reinforcement learning (MARL) methods is that agents reliably execute the actions prescribed by their learned policies. This assumption enables agents to coordinate with others whose actions may not be directly observable. To support such coordination, many MARL algorithms adopt centralised training with decentralised execution (Rashid et al., 2018; Oliehoek et al., 2016). During training, agents have access to global information, while during execution they act using only local observations. This allows agents to form expectations about the behaviour of others under partial observability.

Despite its effectiveness, this training paradigm renders MARL systems highly vulnerable to agent malfunctions. When an agent deviates from its intended behaviour due to faults or failures, the assumptions underpinning coordination can break down, leading to severe performance degradation. Such malfunctions are commonplace in real-world systems (Cristian, 1991). For example, in industrial settings, robots are often required to coordinate closely with other agents, and failures to anticipate correct actions can prevent task completion or lead to dangerous outcomes. Similar challenges arise in settings where control of a single system is factorised across multiple agents, as in multi-agent MuJoCo or dexterous manipulation benchmarks, where each agent controls a distinct subcomponent of the overall system. These considerations motivate the need for MARL methods that remain effective in the presence of agent malfunctions.

In single-agent reinforcement learning (RL), a substantial body of work has studied fault-tolerant (FT) policies that maintain performance under failures (Mguni, 2019; Fan et al., 2021). A common approach, inspired by control theory, introduces an adversarial agent that selects actions to minimise the learner's expected return (Pinto et al., 2017). By exposing the agent to worst-case outcomes, such methods aim to induce robust policies. This formulation leads to a zero-sum game between the controller and the adversary, which is amenable to theoretical analysis due to its structural properties (Osborne & Rubinstein, 1994). However, adversaries that act in exact opposition to the agent's objective often induce overly pessimistic behaviour, causing agents to proceed with excessive caution and degrading performance at higher levels of fault tolerance (Grau-Moya et al., 2018).

To address these challenges, we introduce MARTA, a FT

[1]Queen Mary University London  [2]Peking University  [3]University of Liverpool  [4]Snowflake Inc. Correspondence to: David Mguni <d.mguni@qmul.ac.uk>.

*Proceedings of the 43rd International Conference on Machine Learning*, Seoul, South Korea. PMLR 306, 2026. Copyright 2026 by the author(s).

MARL framework designed to train agents that can robustly respond to agent malfunctions. We focus on cooperative, team-reward settings in which agents may suffer failures and cease to follow the policies they had learned to perform the task. MARTA augments the MARL system with an adaptive agent, termed the Switcher, which observes joint agent behaviour during training and learns when and which agent should malfunction so as to maximally disrupt coordination. Malfunctions are executed by a dedicated Adversary, while the cooperative agents learn policies that best respond to these selectively induced failures.

To enable selective and state-dependent fault induction, MARTA equips the Switcher with switching controls (Øksendal & Sulem, 2007). At each state, the Switcher decides whether to activate a malfunction and, if so, which agent should be affected. Each activation incurs a cost or is constrained by a fixed budget, encouraging malfunctions only in states where they cause a substantial reduction in expected system performance. This design allows MARTA to calibrate the trade-off between robustness and nominal performance, avoiding the conservatism associated with always-on adversarial training. By exploiting information about the joint behaviour of agents, MARTA identifies malfunctions that are most detrimental to coordination and focuses training pressure on these critical situations.

 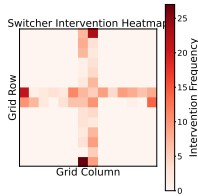

*(a)* Traffic Junction Map     *(b)* Switcher activation heatmap

*Figure 1.* **State-dependent fault induction in Traffic Junction.** (a) shows the Traffic Junction environment, where agents enter from multiple lanes and must coordinate to avoid collisions while clearing the junction. (b) visualises the empirical activation frequency of the learned Switcher; darker regions indicate states where MARTA more often induces a malfunction during training. The Switcher does not simply target states with immediate collisions. Instead, it concentrates interventions at coordination-critical states, including junction, entry, and exit regions, where faults can alter traffic timing, delay lane clearance, and propagate downstream to disrupt future coordination.

As shown in Fig. 1, the learned Switcher identifies states with high downstream coordination impact. In Traffic Junction, faults near entry points can perturb the timing of vehicles entering the intersection, while faults near exits can delay lane clearance and affect following agents. This illustrates the key role of MARTA that malfunctions are induced selectively at states where they are most informative for training robust coordination policies.

Introducing the Switcher and Adversary yields a nonzero-sum Markov game (MG) with $N + 2$ agents (Fudenberg & Tirole, 1991). Convergence in general MGs is rare (Yang & Wang, 2020). Under standard assumptions, which we make explicit in Sec. E.1, the switching control mechanism underlying MARTA admits a well-defined optimisation structure with contraction properties. We show the resulting game admits a unique minimax value and the induced learning dynamics converge under standard assumptions.

**Contributions.**
**1.** We introduce MARTA, a framework for FT MARL based on a Switcher–Adversary mechanism that selectively induces agent malfunctions in coordination-critical states.
**2.** We formulate a fault-switching MG with state-dependent, adversarially selected malfunctions, and prove existence and uniqueness of the minimax value together with convergence of Q-learning procedures under tabular and linear function approximation and malfunction budgets.
**3.** We show MARTA can be implemented as a plug-and-play robustness layer on top of standard MARL algorithms such as QMIX and VDN, without modifying their architectures.
**4.** We evaluate MARTA on Traffic Junction, Level-Based Foraging, and MPE SimpleTag, demonstrating consistent robustness gains and reduced failure rates across discrete and continuous control tasks, and across both Uniform and Worst-case malfunction regimes.

These results establish MARTA as a principled and practical approach to training MARL policies that are robust to realistic agent malfunctions. Unlike classical zero-sum MGs, MARTA incorporates budgeted, state-dependent fault switching, enabling robustness without excessive conservatism.

## 2. The MARTA Framework

We consider a decentralised Markov decision process (Dec-MDP) with agent set $\mathcal{N} = \{1, \ldots, N\}$, global state space $\mathcal{S}$, and agent-specific action spaces $\mathcal{A}^i$. At each timestep, agents simultaneously select actions $\boldsymbol{a} = (a^1, \ldots, a^N)$ and receive a shared team reward. Formally, the Dec-MDP is defined as the tuple $\mathfrak{M} = \langle \mathcal{N}, \mathcal{S}, (\mathcal{A}_i)_{i \in \mathcal{N}}, P, \mathcal{R}, \gamma \rangle$, where $\mathcal{S}$ is a finite set of states, $\mathcal{A}_i$ denotes the action space of agent $i \in \mathcal{N}$, and $\boldsymbol{\mathcal{A}} := \times_{i=1}^{N} \mathcal{A}_i$ is the joint action space. The transition kernel $P : \mathcal{S} \times \boldsymbol{\mathcal{A}} \times \mathcal{S} \to [0, 1]$ governs the system dynamics, and $\gamma \in [0, 1)$ is the discount factor. The reward function $\mathcal{R} : \mathcal{S} \times \boldsymbol{\mathcal{A}} \to \mathcal{P}(D)$ maps state–action pairs to a distribution over rewards, where $D \subset \mathbb{R}$ is compact, and is shared by all agents. We consider a partially observable setting. Given the system state $s_t \in \mathcal{S}$, each agent $i \in \mathcal{N}$ receives a local observation $\tau_i^t = \mathcal{O}(s_t, i)$, where $\mathcal{O} : \mathcal{S} \times \mathcal{N} \to \mathcal{Z}_i$ and $\mathcal{Z}_i$ denotes the observation space of agent $i$. Each agent samples actions according to a Markov policy $\pi_{i, \boldsymbol{\theta}_i} : \mathcal{Z}_i \times \mathcal{A}_i \to [0, 1]$, parameterised by $\boldsymbol{\theta}_i \in \mathbb{R}^d$, with $\pi_i \in \Pi_i$, where $\Pi_i$ is a compact policy space. For

brevity, we write $\pi_i := \pi_{i,\boldsymbol{\theta}_i}$ and denote the joint policy by $\boldsymbol{\pi} = (\pi^1, \ldots, \pi^N) \in \boldsymbol{\Pi} := \times_{i \in \mathcal{N}} \Pi_i$. At time $t$, the system is in state $s_t \in \mathcal{S}$ and each agent selects an action $a_t^i \in \mathcal{A}_i$, forming the joint action $\boldsymbol{a}_t \in \boldsymbol{\mathcal{A}}$. The system then transitions to the next state according to $P$, and each agent receives a reward $r_i \sim \mathcal{R}(s_t, \boldsymbol{a}_t)$. Each agent seeks to maximise the expected discounted return or value function

$$v(s \mid \boldsymbol{\pi}) = \mathbb{E}_{\boldsymbol{\pi}} \left[ \sum_{t=0}^{\infty} \gamma^t \mathcal{R}(s_t, \boldsymbol{a}_t) \,\middle|\, s_0 = s \right]. \quad (1)$$

MARTA includes an adaptive RL agent, termed the Switcher, which observes the joint behaviour of agents and learns when and which agent should malfunction in order to maximally disrupt coordination. Specifically, Switcher selects an agent to malfunction at which point, the agent's actions are decided by an Adversary agent's policy while the other agents execute their intended policy. In response, during training, the remaining agents learn how to respond to the behaviour of the malfunctioning agent and in so doing, learn how to respond to agent malfunctions within the collective. A key feature is that Switcher and Adversary are equipped with their own objective that aims to produce malfunctions that inflict the greatest possible harm to the system (during training) including faults that undermine coordination required to solve the task.

We consider two malfunction regimes: a random fault policy modelling stochastic actuator or sensor failures, and a worst-case policy trained to maximally degrade collective performance. The Switcher learns when and which agent should be affected, enabling the base agents to learn best responses to both mild and catastrophic failures.

In MARTA, the nominal joint action $\boldsymbol{a} \in \boldsymbol{\mathcal{A}}$ is proposed by the agents' policies $\boldsymbol{\pi}$, while a faulty joint action $f \in \boldsymbol{\mathcal{A}}$ is proposed by the adversarial policies $\boldsymbol{\sigma}$. These joint actions are then observed by the policy $\mathfrak{g}$. Switcher samples a discrete decision $g_t \in \mathcal{N}$ from its policy $\mathfrak{g}$. If $g_t = i \in \mathcal{N}$, then agent $i$ is selected to malfunction whereby its action is overridden by an Adversary policy $\sigma^i \in \Pi^i$ while the agents $-i$ sample their actions from their intended policy $\pi^{-i}$ i.e. the joint action $\boldsymbol{a}_t = (f_t^i, a_t^{-i}) \sim (\sigma^i, \pi^{-i})$ is executed in the environment. During training in MARTA, the $N$ agents seek to jointly maximise the following objective: $v(s|\boldsymbol{\pi}, \mathfrak{g}, \boldsymbol{\sigma}) = \mathbb{E}_{\boldsymbol{\pi}, \mathfrak{g}, \boldsymbol{\sigma}} [\sum_{t=0}^{\infty} \gamma^t \mathcal{R}(s_t, \boldsymbol{a}_t)|s = s_0]$. The Adversary and Switcher's objectives are $-v$ which captures their goal to find malfunctions that have the potential to induce the greatest harm to the system. Therefore, by learning an optimal $\mathfrak{g}$, Switcher acquires the optimal policy for activating Adversary. We later show in Theorem D.1 that in response, the MARL agents in turn learn how to best-respond to such failures. As remarked earlier, we consider two scenarios for a malfunctioning agent $i$. The first scenario is when the Adversary action is sampled from a purely random policy and secondly, a worst-case action scenario.

**Switching Control Mechanism.** MARTA allows Switcher to choose, at any state, both which agent malfunctions and whether to activate a malfunction at all. To this end, Switcher is equipped with *switching controls*: at state $s$, using its policy $\mathfrak{g}$, the Switcher selects an action from the discrete set $\mathcal{A}_S = \{0\} \cup \mathcal{N}$, where action $0$ corresponds to no malfunction and action $i \in \mathcal{N}$ triggers a malfunction of agent $i$. If a malfunction is induced for agent $i$ the agent is forced to execute an action using its adversarial policy $\sigma^i$. To encourage selectivity, each activation incurs a positive cost $c > 0$, ensuring that malfunctions are only triggered when they cause a substantial reduction in expected performance. The task of Switcher is therefore to learn a policy $\mathfrak{g}$ that activates adversarial malfunctions solely at states where they most degrade performance. For a given $\boldsymbol{\pi} \in \boldsymbol{\Pi}$, Switcher's objective is to find $\mathfrak{g}$ that *maximises*:[1]

$$v_S(s|\boldsymbol{\pi}, \mathfrak{g}, \boldsymbol{\sigma}) = -\mathbb{E}_{\boldsymbol{\pi}, \mathfrak{g}, \boldsymbol{\sigma}} \left[ \sum_{t=0}^{\infty} \gamma^t \left[ \mathcal{R}(s_t, \boldsymbol{a}_t) + c\mathbf{1}_{\mathcal{N}}(g(s_t)) \right] \right],$$

where $\mathbf{1}_{\mathcal{N}}(g) = 1$ if $g \in \mathcal{N}$ and $\mathbf{1}_{\mathcal{N}}(g) = 0$ otherwise. Switcher's action-value function is $Q_S(s, g|\boldsymbol{\pi}, \mathfrak{g}) = -\mathbb{E}_{\boldsymbol{\pi}, \mathfrak{g}} [\sum_{t=0}^{\infty} \gamma^t (\mathcal{R}(s_t, \boldsymbol{a}_t) + c \cdot \mathbf{1}_{\mathcal{N}}(g))|s_0 = s, g_0 = g]$. Adding the Switcher with an objective distinct from the $N$ agents results in a non-cooperative MG. Having multiple learners with a payoff structure that is neither zero-sum nor a team game can occasion convergence issues (Shoham & Leyton-Brown, 2008). Moreover, unlike standard MARL, MARTA incorporates switching controls. Nevertheless, we prove MARTA converges under standard assumptions.

The parameter $c$ plays an important role in calibrating the fault-tolerance of the system of MARL agents. Larger values of $c$ incur higher costs for each malfunction making Switcher more selective about inflicting malfunctions i.e. limiting interventions to the states where the harm is greatest. This behaviour is consistent with the switching-cost objective and the equilibrium properties established in Theorem 3.1. In turn, the agents learn how to best-respond to only the most harmful malfunctions. When $c \to 0$, we return to a classic robust framework where Switcher can profitably choose to inflict malfunctions at all states, leading to highly cautious joint policies which may harm performance for a given higher fault-tolerance. In Sec. 4, as an alternative to using the cost $c$, we study a setup with a budget constraint on the number of malfunctions Switcher can inflict.

**Details on Architecture.** We now describe a concrete realisation of MARTA's core components which consist of $N$ MARL agents, a MARL agent Adversary and a switching

---

[1] We have employed the shorthand $\boldsymbol{a}_t \sim (\boldsymbol{\pi}, \boldsymbol{\sigma})$ to denote $\boldsymbol{a}_t = (a_t^j, a_t^{-j}) \sim \boldsymbol{\pi}$ when $g_t = 0$ and $\boldsymbol{a}_t = (f_t^j, a_t^{-j}) \sim (\sigma^j, \pi^{-j})$ when $g_t = j \in \mathcal{N}$ i.e. $v_S(s|\boldsymbol{\pi}, \mathfrak{g}, \boldsymbol{\sigma}) = -\mathbb{E}_{\boldsymbol{\pi}, \mathfrak{g}, \boldsymbol{\sigma}} [\sum_{t=0}^{\infty} \sum_{j \in \mathcal{N}} \gamma^t [(\mathcal{R}(s_t, (f_t^j, a_t^{-j})) + c) \mathbf{1}_{(g(s_t)=j)} + \mathcal{R}(s_t, \boldsymbol{a}_t)(1 - \mathbf{1}_{(g(s_t)=j)})]]$. For convenience we drop the dependence of $v_S$ on $\boldsymbol{\sigma}$.

control RL algorithm as Switcher. Each (MA)RL component can be replaced by various other (MA)RL algorithms. MARTA consists of N MARL agents, an Adversary, and a Switcher. Each agent maintains an action policy $\pi_i$ and an adversarial policy $\sigma_i$, both implemented using the same backbone (QMIX or VDN). The Switcher is trained using soft actor–critic with an action space of size $N + 1$, corresponding to no malfunction or activating a malfunction for agent $i$. Training proceeds jointly using a shared replay buffer. Further details are in the Appendix.

## 3. Analysis of MARTA

We establish the existence of a stable equilibrium in which each agent follows a policy that best responds to the system under worst-case malfunctions induced by the Switcher.[2] We further show that the MARTA learning algorithm converges to this equilibrium, which jointly maximises the Switcher's value function and the agents' collective objective. All results in this section are proved in the Appendix and are derived under Assumptions 1–3 (Appendix E.1), which are standard in RL (Bertsekas, 2012).

Although MARTA is formulated as an MG, its structure departs fundamentally from classical zero-sum formulations (Littman, 1994). The induced fault-switching game features: (i) state-dependent fault activation, (ii) persistent mode-switching dynamics, (iii) explicit malfunction budgets, and (iv) a switching-augmented Bellman operator that incorporates activation penalties. The contraction property established in Lemmas E.9–E.12 applies specifically to this operator and does not follow from standard MG results, yielding convergence guarantees not captured by existing adversarial learning frameworks.

The theoretical analysis considers an idealised formulation of MARTA under standard assumptions from stochastic approximation and MG theory, including tabular or linear function approximation. While these guarantees do not extend directly to deep neural network parameterisations, they formally characterise the optimisation problem solved by MARTA and justify the design of the Switcher as a selective fault-induction mechanism. The empirical results in Section 5 evaluate the effectiveness of this mechanism when instantiated with modern deep MARL algorithms.

The learning problem induced by MARTA can be viewed as a nonzero-sum MG $\mathcal{G}$ involving three components: the $N$ cooperative agents, the Switcher, and the Adversary. In the analysis that follows, the Adversary's best response is incorporated into the Switcher's objective, yielding an equivalent reduced game between the agents and the Switcher.

We first observe that an optimal robust policy is one in which the $N$ agents play a best-response joint policy against Switcher who in turn executes a best-response policy. The following result establishes the existence and uniqueness of such a solution in which each agent enacts a best-response i.e. responds optimally to actions of other agents in $\mathcal{G}$. Throughout the analysis, we assume that the Adversary plays a best response to the current policies of the agents and the Switcher. For notational clarity, explicit dependence on the Adversary policy is suppressed.

**Theorem 3.1.** *The minimax value of $\mathcal{G}$ exists and is unique, i.e. there exists a function $v^* : \mathcal{S} \to \mathbb{R}$ such that $v^*(s) := \min_{\hat{\mathfrak{g}}} \max_{\hat{\boldsymbol{\pi}} \in \boldsymbol{\Pi}} v(s|\hat{\boldsymbol{\pi}}, \hat{\mathfrak{g}}) = \max_{\hat{\boldsymbol{\pi}} \in \boldsymbol{\Pi}} \min_{\hat{\mathfrak{g}}} v(s|\hat{\boldsymbol{\pi}}, \hat{\mathfrak{g}}), \quad \forall s \in \mathcal{S}.$*

*Sketch.* The minimax equality and existence of $v^*$ follow from the contraction property of the induced Bellman operator. Lemma 7 establishes monotonicity and Lemma 8 proves strict contraction under Assumptions 1–3 (Appendix D.1), implying a unique fixed point by the Banach fixed-point theorem. This fixed point coincides with the unique minimax value of the game. $\square$

**Proposition 3.2.** *Let $\hat{\boldsymbol{\pi}} \in \boldsymbol{\Pi}$ be an equilibrium policy as defined in Theorem 3.1, then $\hat{\boldsymbol{\pi}}$ is a Markov perfect equilibrium[3] policy, that is to say, no agent can improve the team reward by responding to the malfunctions executed by Adversary with a change in their policy.*

This follows from the fact that all players optimise stationary objectives under Markovian dynamics, so no history-dependent deviation can improve the equilibrium outcome.

Having established the existence of a solution to the game, we now turn to the question of how a set of MARL agents can learn such a solution. Theorem D.1 (see Sec. E in Appendix) proves the convergence of a dynamic programming procedure to the solution. It therefore lays the foundation for our MARL algorithm, MARTA which learns robust MARL policies in this setting. In particular, the following theorem proves the convergence of a MARTA to the solution $v^*$ by repeated application of a Bellman operator. A direct consequence of the equilibrium structure and the switching-cost objective is that optimal switching policies activate malfunctions only when the expected degradation in performance outweighs the associated cost.

**Theorem 3.3.** *Consider the following Q-learning variant:*

$$\boldsymbol{Q}_{t+1}(s_t, \boldsymbol{a}_t, g) = \boldsymbol{Q}_t(s_t, \boldsymbol{a}_t, g)$$
$$+ \alpha_t(s_t, \boldsymbol{a}_t) \big[ \min \big\{ \hat{\boldsymbol{\mathcal{M}}} \boldsymbol{Q}_t(s_t, \boldsymbol{a}_t, g), \mathcal{R}_S(s, \boldsymbol{a}, g)$$
$$+ \gamma \max_{\boldsymbol{a}' \in \boldsymbol{\mathcal{A}}} \boldsymbol{Q}_t(s_{t+1}, \boldsymbol{a}', g) \big\} - \boldsymbol{Q}_t(s_t, \boldsymbol{a}_t, g) \big],$$

---

[2]Since the Adversary uses a stochastic policy, the limiting variance case covers scenarios in which a malfunctioning agent executes random actions.

[3]$(\pi^i, \pi^{-i}) = \boldsymbol{\pi} \in \boldsymbol{\Pi}$ is a Markov perfect equilibrium if no agent can improve the expected return by changing their policy i.e. $\forall i \in \mathcal{N}, \forall \pi'^i \in \Pi^i$, we have $v(s|\boldsymbol{\pi}, \mathfrak{g}) - v(s|(\pi'^i, \pi^{-i}), \mathfrak{g}) \leq 0.$

*then $Q_t$ converges to $Q^\star$ with probability 1, where $s_t, s_{t+1} \in \mathcal{S}$ and $a_t \in \mathcal{A}$.*

Theorem 3.3 proves the convergence of MARTA to the solution to the game $\mathcal{G}$ in which each agent optimally responds to the policies of other agents in the system.

In Prop. E.8, we fully characterise the points where Switcher should activate Adversary and which agent in terms of an obstacle condition which can be determined at each state. We now describe a Q-learning procedure that computes the value of the game. The update extends standard Q-learning by incorporating the Switcher's action into the state–action value function, resulting in a joint Bellman operator that captures both agent actions and switching decisions.

Function approximators enable parameterisation of key estimators in RL. We extend the convergence results to linear function approximation, a standard convex setting and a foundation for more expressive parameterisations.

**Theorem 3.4.** *MARTA converges to the stable point of $\mathcal{G}$, moreover, given a set of linearly independent basis functions $\Phi = \{\phi_1, \ldots, \phi_p\}$ with $\phi_k \in L_2, \forall k$. Define a projection $\Pi$ on a function $\Lambda$ by: $\Pi\Lambda := \underset{\bar{\Lambda} \in \{\Psi r | r \in \mathbb{R}^p\}}{\arg\min} \|\bar{\Lambda} - \Lambda\|$. Then MARTA converges to a limit point $r^\star \in \mathbb{R}^p$ which is the unique solution to $\Pi\mathfrak{F}(\Phi r^\star) = \Phi r^\star$ where for any $\Lambda \in L_2$, $\mathfrak{F}\Lambda := \mathcal{R} + \gamma P \min\{\hat{\mathcal{M}}\Lambda, \Lambda\}$ and $r^\star$ satisfies the following: $\|\Phi r^\star - Q^\star\| \leq (1 - \gamma^2)^{-1/2} \|\Pi Q^\star - Q^\star\|$.*

Theorem 3.4 shows that, under linear function approximation, the projected Bellman operator associated with the switching control mechanism remains a contraction. The projection step ensures stability of the update within the feature space, while the contraction guarantees convergence to a unique fixed point in parameter space.

## 4. MARTA with a Malfunction Budget

The fault-activation cost plays a critical role in ensuring that the Adversary induces malfunctions which cause a significant reduction in the system performance. Despite the intuitive interpretation, in some applications, the choice of $c$ is not obvious. We now consider an alternative view and introduce a variant of MARTA, namely MARTA-B that imposes a budgetary constraint of the number of malfunctions that can be induced by Adversary during training. We show that by tracking its remaining budget, MARTA-B is able to learn a Switcher policy that makes optimal usage of its budget while respecting the budget constraint almost surely. The Switcher now has the following constrained problem:

$$\min_{\hat{\mathfrak{g}}} \max_{\hat{\pi} \in \Pi} v_S(s | \hat{\pi}, \hat{\mathfrak{g}}) \text{ s. t. } n - \sum_{k < \infty} \sum_{t_k \geq 0} \mathbf{1}(\mathfrak{g}(\cdot | s_{t_k})) \geq 0,$$

where $n \geq 0$ is a fixed value that represents the budget for the number of malfunctions and the index $k = 1, \ldots$

represents the training episode count. As in (Sootla et al., 2022; Mguni et al., 2023), we introduce a new variable $y_t$ that tracks the remaining number of activations: $y_t := n - \sum_{t \geq 0} \mathbf{1}(\mathfrak{g}(s_t))$ where the variable $y_t$ is now treated as the new state variable which is a component in an augmented state space $\mathcal{X} := \mathcal{S} \times \mathbb{N}$. In Theorem D.2, we prove the convergence of MARTA-B i.e., MARTA with a fixed Switcher budget to the optimal joint value function.

## 5. Experiments

We conduct a series of experiments to verify: **(1) Plug-and-play effectiveness.** MARTA can be attached to standard MARL backbones without architectural changes and consistently improves robustness under injected malfunctions; **(2) Mechanism validity and calibration.** the learned *Switcher* meaningfully decides *when* and *whom* to intervene on.

**Environments**

**(1) Traffic Junction (TJ)** (Koul, 2019). A discrete gridworld in which agents must navigate through intersections without collisions. This environment emphasises high-density spatial conflict resolution and tests the agents' ability to handle localised, time-critical disruptions.

**(2) Level-Based Foraging (LBF)** (Christianos et al., 2020). Agents control units of a certain level, food items of varying levels are scattered across the map, and each agent aims to collect as much as possible. An agent can only collect food if the cumulative level of the adjacent agents performing the 'collectqaction is greater than or equal to the food's level. It captures the spectrum between independent behaviour and tightly coupled cooperation under partial observability.

**(3) Multi-Agent Particle Environment (MPE): Simple-Tag** (Lowe et al., 2017). A continuous control domain where multiple pursuers must coordinate to capture the evader. It emphasises coordination under non-stationarity. Individual agents not coordinating can severely reduce rewards.

**(4) StarCraft Multi-Agent Challenge v2 (SMACv2).** To evaluate MARTA at larger scale and under more challenging partial observability, we additionally benchmark on SMACv2. We evaluate MARTA using QMIX as the base learner and report win rate and episode return. Faults are injected using the same mechanisms as in earlier experiments, including Uniform and Worst-case fault policies, fixed and resampled faulty agents, and aligned versus shifted train-test fault distributions. We additionally report fault-conditioned win rate, defined as the win rate conditioned on at least one malfunction occurring during the episode. This setting tests whether MARTA improves robustness to agent malfunctions without degrading nominal task performance.

**Experiment Configurations.** To make the robustness setting transparent, we factor our design into a few orthogonal

axes. Each run is determined by: During MARTA training, malfunctions are selected by the learned Switcher: at each state, the Switcher decides whether to activate a malfunction and, if so, which agent should be affected. Thus, no exogenous per-step malfunction probability is used to trigger MARTA training faults. During evaluation, however, all methods are tested under a controlled external malfunction process. At each timestep, a malfunction may occur with probability $p$; when it occurs, the affected agent is selected according to the evaluation fault distribution $F$, and its action is replaced by a corrupted policy $\sigma^i$. This evaluation protocol allows us to compare MARTA and non-MARTA baselines under the same test-time fault regimes. Each evaluation run is determined by: *who* malfunctions, specified by the agent-selection distribution $F$; *how* the malfunction behaves, specified by the fault policy $\sigma^i$; the per-step evaluation trigger probability $p$; the switch cost $c$ used during MARTA training to calibrate interventions; and whether the train–test fault processes are aligned or shifted. Table 2 summarises these factors.

To facilitate analysis, we categorise our experiments into three difficulty bands. **Easy (Level 1)** evaluates aligned and predictable faults, where the same fixed agent may malfunction at test time with probability $p$. **Medium (Level 2)** evaluates aligned but dynamic faults, where the malfunctioning agent is resampled from $F$ over time. **Hard (Level 3)** evaluates train–test shift, where the test-time fault distribution, malfunction probability, or fault policy differs from the process encountered during training. Together these bands provide a structured view of robustness, ranging from predictable aligned malfunctions to adversarial distribution shifts with multiple concurrent faults.

**Evaluation Protocol.** Every $10,000$ steps, agents are evaluated for more than $100$ episodes using the same configuration of the malfunction. The best-performing checkpoint is reported based on average return. Unless otherwise specified, all hyperparameters and architecture configurations are identical across MARTA and its baseline counterpart. In all plots **dark lines represent averages over 3 seeds and the shaded regions represent $95\%$ confidence intervals**. We summarise where key experimental details are located: *Environment configurations, agent counts and fault parameters* are in Table 3 in the Appendix. *Train–test fault protocols (aligned, shifted and resampled)* are summarised in Table 2. Agent counts are: TJ (4–6 agents), LBF (4 agents), and MPE SimpleTag (4 agents). *Detailed descriptions of how faults are generated and controlled* are in Table 2.

### Can MARTA improve MARL algorithm performance and reduce failure modes?

To validate our claim that MARTA enhances robustness and improves performance, we evaluate MARTA on TJ, LBF

(5x5-4p-1f), and MPE (SimpleTag) under injected malfunctions. In each environment, we compare a base MARL algorithm with its MARTA-augmented counterpart. As shown in Fig. 2, MARTA-QMIX achieves final return gains of **11.1%** in TJ, **21.4%** in MPE, **44.6%** in LBF and **114.9%** and **9.3%** in SMACv2, 3m and 8m respectively. Complete results are reported in Table 4 in Sec. B. Robustness is particularly difficult in smaller-agent settings, where the failure of a single agent has a disproportionate impact on coordination. This effect is evident in TJ and LBF, where limited redundancy magnifies the influence of faults. Despite this, MARTA delivers consistent improvements, including in MPE, where continuous control and adversarial dynamics further increase task complexity. Hence, MARTA enhances fault tolerance across a diverse range of tasks without changes to the base learner's architecture. In addition, **MARTA enables the baseline algorithm to avoid the dangers** of agent collisions in the TJ environment. In TJ we define the failure rate as the episode-level collision rate: $\text{TJ\_collision\_rate} = \frac{1}{M} \sum_{e=1}^{M} \mathbf{1}\{\text{episode } e \text{ contains any collision}\}$ where $M$ is the total number of episodes. We plotted the failure rates of each algorithm in Fig. 3a, which shows that MARTA significantly reduces failure rates.

**The importance of Switching Controls.** A key component of MARTA is the switching control mechanism. This enables the Switcher to choose to activate malfunctions at high-risk states where learning robust behaviour is critical. To evaluate the impact of switching controls, we compared MARTA with a variant of MARTA in which the Switcher's policy is replaced with a Bernoulli random trigger: for some $p \in (0, 1]$, at each state, a malfunction for each agent occurs with probability $p/|\mathcal{N}|$, and with probability $1 - p$ no malfunction is activated. Fig. 3b shows incorporating the ability to learn an optimal switching control yields much better performance compared to activating malfunctions randomly ("random policy").

**Experiment parameters.** In Sec. A, we also perform ablations on the switch cost $c$ which reveals that adjusting this parameter increases the system's preference for FT in the fault–performance trade-off as $c \downarrow 0$. We further ablate the malfunction probability $p$ and the agent count $N$, demonstrating the robustness of MARTA to these parameters.

**MARTA is a plug-and-play enhancement framework.** To validate our claim that MARTA easily adopts MARL algorithms, we tested the improvements MARTA delivers with an alternative base MARL algorithm. Specifically, we replaced QMIX in MARTA with VDN (Sunehag et al., 2018). Fig. 4 shows learning curves. In the TJ environment, MARTA-VDN yields a **37.8%** gain over its VDN baseline and in SMACv2 3m, MARTA-VDN yields a **116.7%** gain over its VDN baseline. In all maps, MARTA-VDN significantly outperforms the base VDN algorithm.

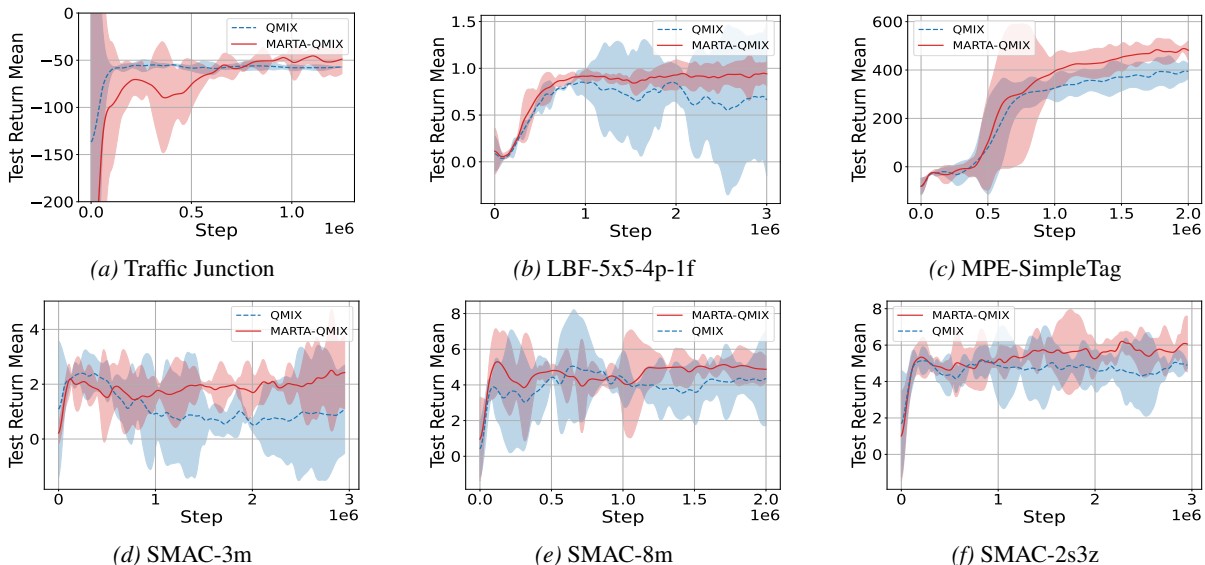

*Figure 2.* **Robustness against malfunctions.** Each plot compares a base MARL algorithm with and without MARTA in faulty agent settings. In all scenarios, MARTA improves robustness.

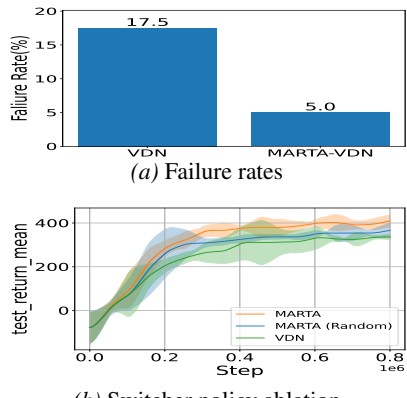

*(a)* Failure rates

*(b)* Switcher policy ablation

*Figure 3.* Ablation results in MPE.

**Comparison with robust adversarial MARL baselines.** We compare against M3DDPG (Lowe et al., 2017) and EIR (Li et al., 2024) as representative robust and adversarial MARL baselines applicable under explicit agent-malfunction processes; other robust MARL methods assume different threat models and are therefore not directly comparable. We adapt M3DDPG as a robust baseline on the MPE SIMPLETAG environment and construct a MARTA-enhanced variant, MARTA-MADDPG, by applying our switching framework to the MADDPG backbone. Figure 5 reports learning curves for MADDPG, M3DDPG, and MARTA-MADDPG under the same malfunction process. MARTA-MADDPG consistently outperforms both MADDPG and M3DDPG, demonstrating that MARTA provides additional robustness gains even when applied on top of an established adversarial MARL method.

**Robustness under dynamic fault distributions.** Figure 6 compares MARTA against EIR and reports performance under two qualitatively different fault regimes. In Case 1,

malfunctions affect a fixed agent with constant probability, yielding a stable and predictable fault distribution shared between training and testing. In Case 2, faults occur dynamically: at each timestep, any agent may malfunction with fixed probability, causing the location of the fault to switch across agents and time and introducing a pronounced train–test distribution mismatch. The first row (TJ and LBF) reports risk metrics, collision rate and failure rate, where lower values indicate safer and more reliable behaviour. The second row (MPE and SMAC-3m) reports success and coordination metrics, namely capture rate and focused fire rate, where higher values indicate stronger coordination under malfunction. While MARTA performs comparably to EIR under the aligned Case 1 regime, it consistently outperforms EIR under the dynamic Case 2 regime across all environments, indicating stronger robustness and generalisation to multi-location, time-varying faults.

Further experimental results, ablations, and diagnostic analyses are provided in the Appendix, including extended evaluations under alternative fault models, additional visualisations of switching behaviour, and supplementary quantitative results. These studies reinforce the empirical robustness of MARTA across settings and help disentangle the contributions of its individual components.

**Computational overhead.** In all experiments, we reuse the same architectures and hyperparameters as the underlying MARL baselines and train the Switcher with a lightweight actor–critic learner. This introduces only a modest training-time overhead. At execution time there is no auxiliary safety filter or online optimisation so the wall-clock cost of MARTA is close to that of the base learner alone. Table 1 reports wall-clock and memory overhead on SMACv2 en-

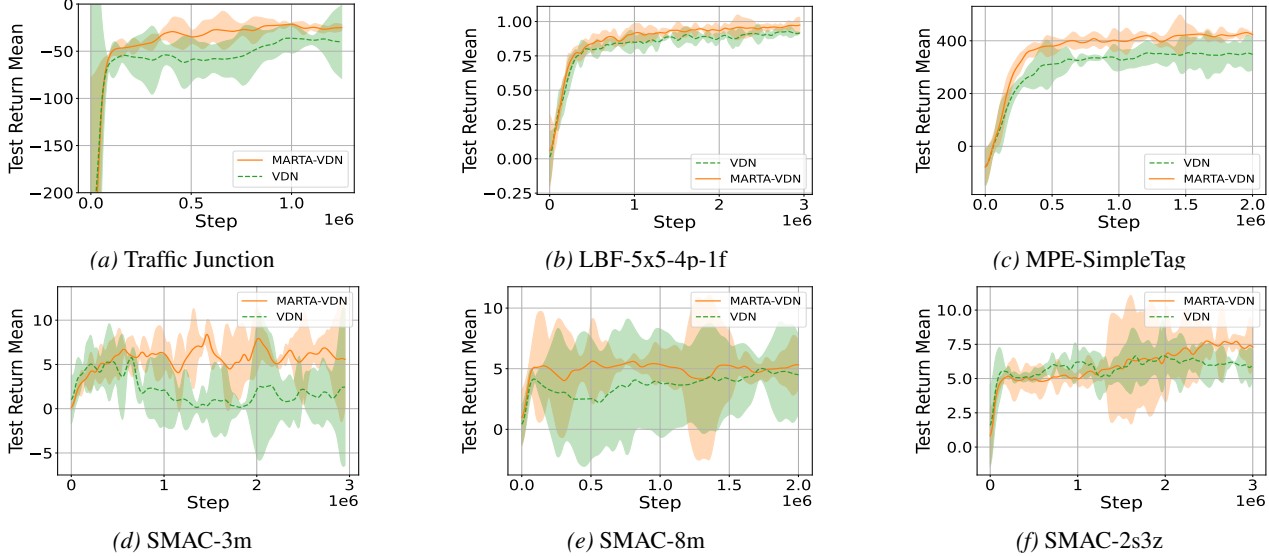

*(a)* Traffic Junction  *(b)* LBF-5x5-4p-1f  *(c)* MPE-SimpleTag

*(d)* SMAC-3m  *(e)* SMAC-8m  *(f)* SMAC-2s3z

*Figure 4.* **Performance of MARTA+VDN.** MARTA improves performance in all scenarios.

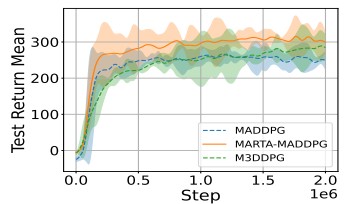

*Figure 5.* Comparison between MADDPG with M3DDPG and MARTA-MADDPG in MPE.

*Table 1.* **Computational overhead of MARTA.** Wall-clock time and GPU memory are reported as multipliers relative to the corresponding base MARL learner. The overhead is incurred only during training; execution uses the learned cooperative policies directly.

| Env. | #Agents | Train time | GPU mem. |
|---|---|---|---|
| 3m | 3 | 1.10× | 1.27× |
| 8m | 8 | 1.15× | 1.43× |
| 10m-vs-11m | 10 | 1.07× | 1.52× |
| 25m | 25 | 1.05× | 1.61× |
| Execution | – | 1.00× | no extra component |

vironments with different numbers of agents. At execution time, the learned cooperative agents act directly, so MARTA introduces no additional execution-time communication or online decision cost.

## 6. Related Work

**Fault tolerance and safety in MARL.** Safety and robustness in MARL have been studied through shielding, backup policies and constrained optimisation. Shielding approaches (Zhang et al., 2019; Elsayed-Aly et al., 2021) use additional safety layers or backup policies to override unsafe actions.

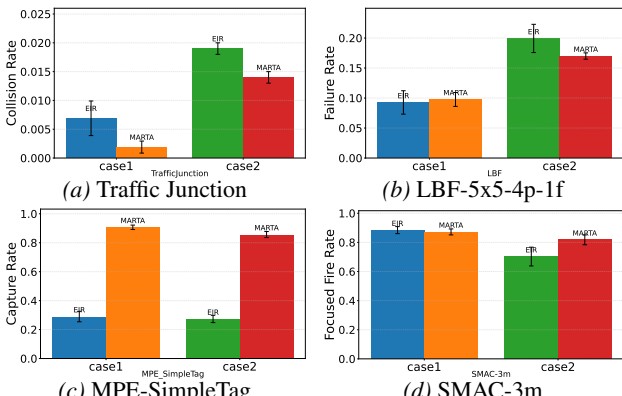

*(a)* Traffic Junction  *(b)* LBF-5x5-4p-1f

*(c)* MPE-SimpleTag  *(d)* SMAC-3m

*Figure 6.* **Evaluation of MARTA and EIR across four environments under two fault regimes.** Case 1 (fixed fault): a single agent fails with fixed probability, with aligned train–test distributions. Case 2 (dynamic random fault): at test time, any agent may malfunction at any timestep, inducing a train–test distribution mismatch. Top row reports risk metrics (lower is better); bottom row reports success and coordination metrics (higher is better).

Qin et al. (Qin et al., 2021) employ control barrier functions to enforce safety constraints, but without formal guarantees. These methods often require per-timestep safety checks or dedicated certificates, and their cost grows with the number of agents. In contrast, MARTA embeds robustness directly into the training dynamics avoiding runtime safety layers and preserving the architecture of the underlying MARL learner. Constrained MARL formulations (Gu et al., 2021; Lu et al., 2021) treat safety as a constrained optimisation problem. These methods often face convergence and scalability challenges. By contrast, the MG underlying MARTA has a unique solution to which MARTA has convergence guarantees for tabular and linearly approximated settings.
**Robust and adversarial MARL.** Adversarial training methods for RL and MARL (Pinto et al., 2017; Li et al., 2019;

Zhang et al., 2020) typically introduce an opponent that perturbs actions, observations or dynamics to construct worst-case trajectories. These methods improve robustness but often induce overly conservative behaviour, since the agent is trained under an adversary that is active at every step. Moreover, most such work focuses on perturbations in a single-agent MDP or on model uncertainty, rather than on explicit agent malfunctions in cooperative teams. MARTA differs in three ways. First, it targets actuator-level failures in which an entire agent temporarily loses control to a fault policy, rather than small perturbations around nominal actions or states. Second, it models the timing and location of faults through a Switcher that explicitly reasons over state-dependent costs or budgets, rather than assuming an always-on adversary. Third, it provides convergence guarantees for this switching-augmented game, including under linear function approximation and budget constraints.

**Diagnostics and poisoning attacks in MARL.** Recent work has examined the vulnerability of MARL systems to targeted perturbations or poisoning attacks. RTCA (Zhou & Liu, 2023) proposes a resilience testing framework that perturbs the states of critical agents to expose weaknesses. Zheng et al. study training-time poisoning in which a single manipulated agent can poison policies. These works are primarily *diagnostic* or *attack-oriented*: they evaluate the weakness of existing MARL policies or design efficient attacks, rather than providing a defence scheme that yields robust policies. MARTA is complementary. It is a training-time defence mechanism that induces controlled, state-dependent malfunctions then trains agents to jointly best respond.

**Byzantine-robust MARL and adversarial teammates.** Li et al. (Li et al., 2024) study Byzantine-robust cooperative MARL through a Bayesian game formulation, in which some teammates may behave adversarially. Their focus is on strategic deviations modelled through adversarial types and on robust reasoning about such behaviour. MARTA instead models non-strategic actuator malfunctions, such as stuck actuators, corrupted control modules and state-dependent controller failures. These represent different robustness regimes. Strategic adversaries may deliberately coordinate to mislead, whereas actuator faults in physical systems are often non-strategic yet catastrophic for coordination. MARTA introduces a specific fault-switching MG (with budgets), and proves existence and uniqueness of a minimax value and convergence under both tabular and linearly approximated settings. This yields a different set of theoretical guarantees tailored to the malfunction setting.

**Shielding, backup policies and scalability.** Shielding and backup-policy methods (Zhang et al., 2019; Elsayed-Aly et al., 2021; Qin et al., 2021) provide valuable tools for enforcing safety constraints by modifying actions at execution time. However, their reliance on online constraint checking and per-agent safety mechanisms can create scalability challenges as the number of agents grows. MARTA takes a complementary approach which avoids runtime safety layers and allows robustness to scale with the underlying MARL learner without redesigning its architecture. Finally, MARTA is intentionally plug-and-play. It attaches to standard value-based and actor–critic MARL algorithms without altering their internal networks, and can also be combined with more sophisticated robust architectures. In this sense, MARTA acts as a general robustness layer that complements rather than replaces existing robust MARL techniques.

## Conclusion

We introduced MARTA, a fault-tolerant framework for cooperative MARL that selectively induces agent malfunctions during training to improve robustness. By formulating fault tolerance as a switching-augmented MG, we established theoretical guarantees for equilibrium existence and convergence under standard assumptions. Empirical results across diverse benchmarks demonstrate that MARTA improves robustness to dynamic and adversarial malfunctions while remaining plug-and-play with existing MARL algorithms. Together, these results position MARTA as a principled and practical approach to fault-tolerant MARL.

## Impact Statement

This paper presents work whose goal is to advance the robustness and reliability of multi-agent reinforcement learning. The proposed methods are intended to improve fault tolerance and safety in multi-agent systems, which may benefit applications such as robotics and distributed control. We do not foresee significant negative societal consequences specific to this work beyond those commonly associated with the deployment of machine learning systems.

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

# Supplementary Material

## A. Ablation Studies

Switcher **Effectiveness and Calibration.** We evaluate the effectiveness of MARTA's Switcher mechanism in two aspects: (1) Its ability to *calibrate* the frequency of adversarial malfunctions via the parameter $c$; (2) Its performance in Level 2 (training/test malfunction distribution shift):

**Varying Switching Cost** $c$ **(Fig. 7a and Fig. 7c).** We train MARTA under different switching costs $c$ in the TJ environment. Larger values of $c$ make Switcher more conservative in triggering malfunctions, resulting in fewer fault activations. This reflects a trade-off: high $c$ limits unnecessary disruptions but may reduce robustness; low $c$ encourages aggressive adversarial training. The monotonic trend validates that Switcher adaptively regulates the fault difficulty during training.

**Is MARTA robust under malfunction distributional shifts? (Fig. 7b).** We compare MARTA under **Case 1** (aligned malfunction distributions in training and testing) and **Case 2** (mismatched distributions), across different switching costs $c$ in the TJ environment. The performance gap between the cases highlights the generalisation difficulty under distributional shift.

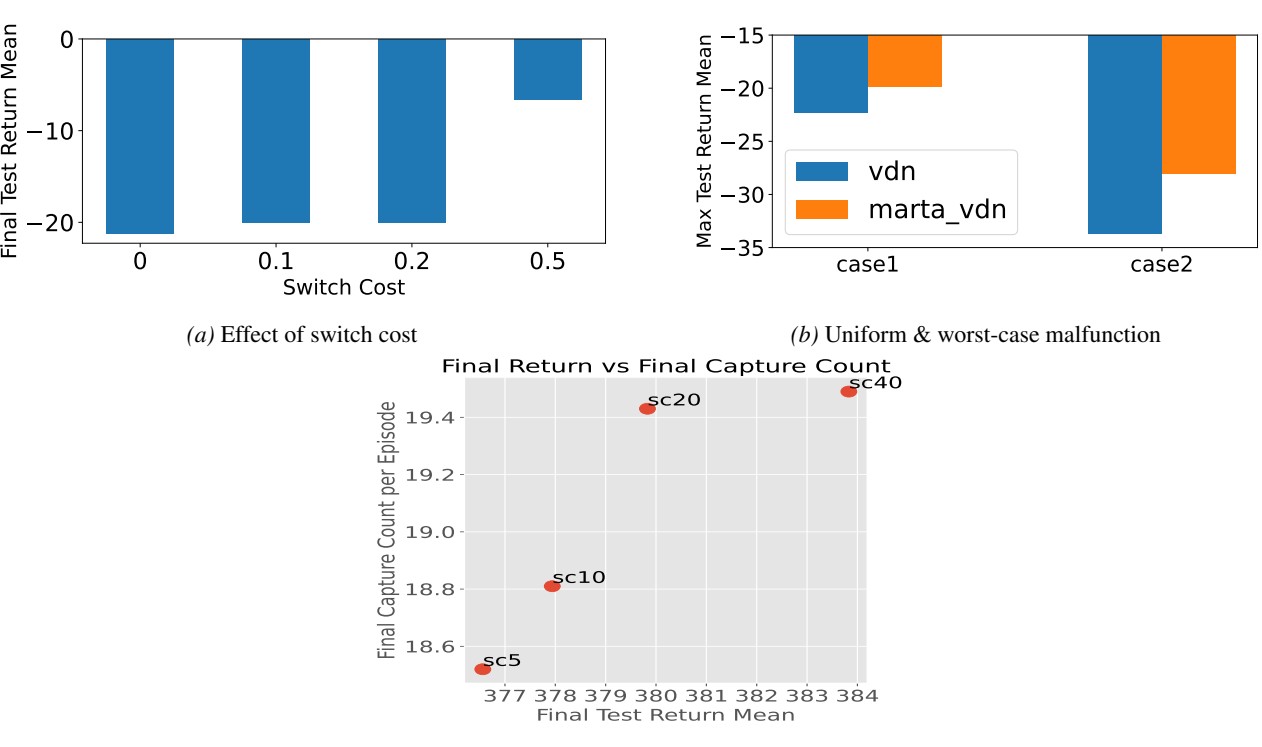

*(a)* Effect of switch cost          *(b)* Uniform & worst-case malfunction

*(c)* Trade-off between return and Collision rate

*Figure 7.* Switcher **ablation experiments.** (a) shows final evaluation returns under different switching costs $c$. (b) compares MARTA in Case 1 and Case 2.

**Is MARTA robust under Varying Malfunction Probability?** We evaluate the fault-tolerance of MARTA across different levels of malfunction probability $p$. Specifically, $p$ denotes the per-timestep probability that a malfunction is introduced i.e., at each timestep, with probability $p$, an agent becomes faulty and executes the malfunction policy. Larger $p$ values indicate more frequent and unpredictable failures, thereby increasing the difficulty of the task. We conduct this experiment in the **Traffic Junction** environment, using QMIX as the base learner. As shown in Fig. 8, we vary $p$ and compare the final returns of MARTA-QMIX with QMIX. The results show a clear performance gap: while the baseline QMIX degrades rapidly under higher malfunction frequencies, MARTA-QMIX does not suffer performance loss even at higher $p$. This suggests the Switcher mechanism in MARTA enables more effective adaptation to dynamic and persistent failure conditions. In summary, MARTA significantly extends the robustness envelope of its underlying MARL algorithm, showing consistent advantage across a wide range of disturbance intensities.

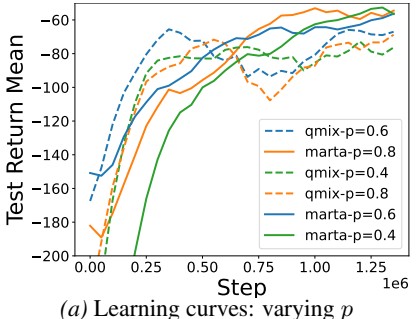

*(a)* Learning curves: varying $p$

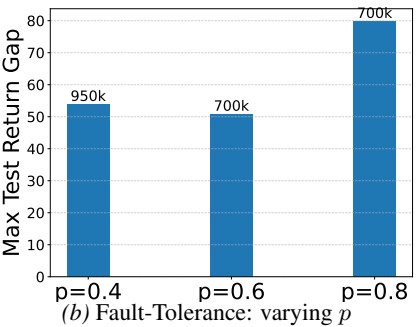

*(b)* Fault-Tolerance: varying $p$

*Figure 8.* Robustness under varying malfunction probability $p$ in the TJ environment.

# B. Further Analysis of Experimental Results

*Table 2.* Experimental factors, levels, and rationale

| Factor | Levels | Rationale & Motivation |
|---|---|---|
| Agent selection (*who*) $F$ | {*Simple, Mid, Hard*} | **Simple**: same fixed agent $i^\star$ fails with per-step prob. $p$ in train/test; persistent, predictable single-point failure. **Mid**: per step, sample one agent $i_t \sim F$; train/test share $F$, unpredictable yet aligned. **Hard**: all agents faultable (concurrent allowed) and train/test fault processes mismatched ($F_{\text{test}} \neq F_{\text{train}}$, $p_{\text{test}} \neq p_{\text{train}}$); strongest generalisation stress. |
| Fault policy (*how*) $\sigma^i$ | {*Uniform, Worst-case*} | **Uniform**: uniformly random action; noise baseline. **Worst-case**: adversarial via $\operatorname{softmax}(-Q^*)$; targets high-impact disruption without altering environment dynamics. |
| Trigger probability $p$ | grid $\mathcal{P}$ | Controls fault frequency (difficulty). Sweeps over $p \in \mathcal{P}$ quantify degradation and intervention–return trade-offs. |
| Switch cost $c$ | grid $\mathcal{C}$ | Calibrates Switcher activation; larger $c$ discourages frequent interventions. Sweeps test robustness–efficiency trade-off. |
| Train–test protocol | {*Aligned, Shifted*} | **Aligned (Case 1)**: test uses same $F, \sigma^i$ as training; ID robustness. **Shifted (Case 2)**: test alters $F$ and/or $\sigma^i$; OOD generalisation. |
| Switcher type (ablation) | {*Learned, Random*} | **Random**: fixed Bernoulli triggers with all else fixed; isolates the benefit of state-dependent, learned switching (Learned). |

*Table 3.* Settings for Fig. 2 and Fig. 4.

| Env | Base Algo (Agents) | Malfunction | $p$ | Fig |
|---|---|---|---|---|
| TJ | VDN (4) | Level2 | 0.1 | a |
| | QMIX (6) | Level1 | 0.1 | a |
| LBF | VDN (4) | Level2 | 0.4 | b |
| | QMIX (4) | Level2 | 0.2 | b |
| MPE | QMIX (4) | Level1 | 0.1 | c |
| | MADDPG (4) | Level2 | 0.4 | c |
| SMAC-3m | VDN (3) | Level1 | 0.4 | d |
| | QMIX (3) | Level2 | 0.2 | d |
| SMAC-8m | VDN (8) | Level1 | 0.2 | d |
| | QMIX (8) | Level2 | 0.2 | d |
| SMAC-2s3z | VDN (5) | Level1 | 0.6 | d |
| | QMIX (5) | Level2 | 0.2 | d |

To provide a quantitative summary, we evaluate each method using two metrics:

**Final Test Return:** We average the test return from the last 5 evaluation checkpoints for each seed, and report the mean ± standard deviation across seeds. As shown in Table 4, MARTA exhibits clear performance gains in all three scenarios. The column "Gain (%)" indicates the relative improvement of MARTA over its baseline, computed as the percentage change with respect to the baseline's absolute value, i.e., (MARTA − Baseline)/|Baseline| × 100.

**Area Under the Learning Curve (AUC):** To jointly evaluate sample efficiency and final return, we compute the AUC over the test return curve for each seed in Fig 9. MARTA consistently achieves higher AUC scores than its baselines, indicating both faster and more stable learning. The AUC "Gain (%)" reflects the relative increase in cumulative performance throughout training. Note that in TRAFFIC JUNCTION, MARTA-QMIX exhibits a lower AUC than QMIX because it converges more slowly in the early phase; however, it ultimately attains a higher final return.

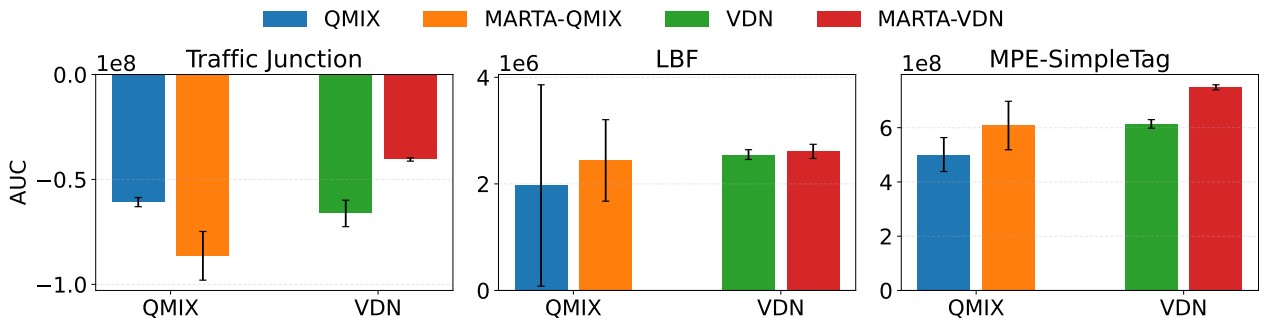

Figure 9. AUC comparisons across environments.

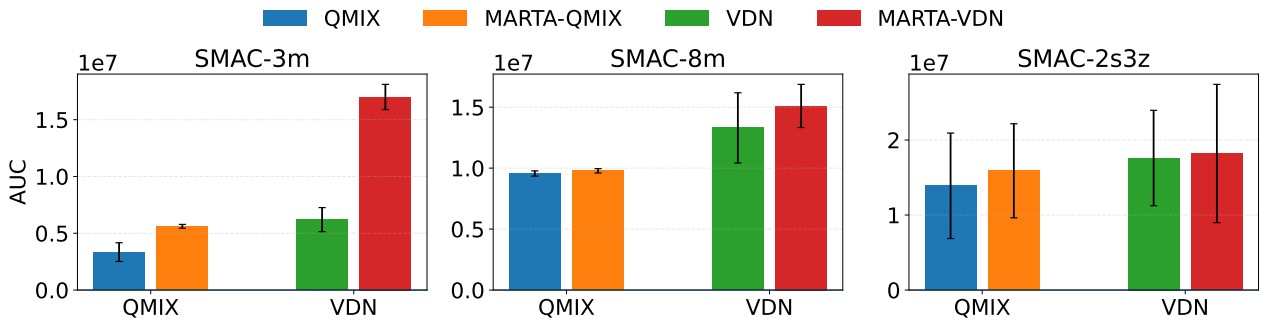

Figure 10. AUC comparisons across smac environments.

Table 4. The improvements of final return and AUC (± std) over baselines across 3 seeds.

| Env | Method | Final Return ($\mu \pm \sigma$) | Gain (%) | AUC ($\mu \pm \sigma$) | Gain (%) |
|---|---|---|---|---|---|
| Traffic Junction | QMIX | $-54.2 \pm 1.1$ | | $-6.08\mathrm{e}7 \pm 2.19\mathrm{e}6$ | |
| | MARTA-QMIX | $-48.2 \pm 4.7$ | +11.1 | $-8.64\mathrm{e}7 \pm 11.6\mathrm{e}6$ | -42.1 |
| | VDN | $-40.1 \pm 2.2$ | | $-6.62\mathrm{e}7 \pm 6.31\mathrm{e}6$ | |
| | MARTA-VDN | $-24.9 \pm 0.8$ | +37.8 | $-4.05\mathrm{e}7 \pm 0.79\mathrm{e}6$ | +38.8 |
| LBF | QMIX | $0.65 \pm 0.279$ | | $1.97\mathrm{e}6 \pm 18.9\mathrm{e}5$ | |
| | MARTA-QMIX | $0.94 \pm 0.043$ | +44.6 | $2.44\mathrm{e}6 \pm 7.65\mathrm{e}5$ | +23.9 |
| | VDN | $0.94 \pm 0.002$ | | $2.55\mathrm{e}6 \pm 0.92\mathrm{e}5$ | |
| | MARTA-VDN | $0.97 \pm 0.021$ | +3.2 | $2.61\mathrm{e}6 \pm 1.33\mathrm{e}5$ | +2.4 |
| MPE-SimpleTag | QMIX | $395.28 \pm 14.78$ | | $5.01\mathrm{e}8 \pm 6.26\mathrm{e}7$ | |
| | MARTA-QMIX | $479.79 \pm 26.19$ | +21.4 | $6.08\mathrm{e}8 \pm 8.94\mathrm{e}7$ | +21.4 |
| | VDN | $346.80 \pm 19.01$ | | $6.14\mathrm{e}8 \pm 1.58\mathrm{e}7$ | |
| | MARTA-VDN | $420.26 \pm 5.789$ | +21.2 | $7.49\mathrm{e}8 \pm 0.92\mathrm{e}7$ | +22.0 |
| SMAC-3m | QMIX | $1.14 \pm 0.53$ | | $3.34\mathrm{e}6 \pm 8.24\mathrm{e}5$ | |
| | MARTA-QMIX | $2.45 \pm 0.46$ | +114.9 | $5.61\mathrm{e}6 \pm 1.70\mathrm{e}5$ | +68.0 |
| | VDN | $2.57 \pm 3.14$ | | $0.62\mathrm{e}7 \pm 1.06\mathrm{e}6$ | |
| | MARTA-VDN | $5.57 \pm 1.74$ | +116.7 | $1.70\mathrm{e}7 \pm 1.11\mathrm{e}6$ | +22.0 |
| SMAC-8m | QMIX | $4.18 \pm 0.36$ | | $9.56\mathrm{e}6 \pm 2.16\mathrm{e}5$ | |
| | MARTA-QMIX | $4.57 \pm 0.45$ | +9.3 | $9.78\mathrm{e}6 \pm 1.78\mathrm{e}5$ | +2.3 |
| | VDN | $5.08 \pm 0.38$ | | $1.33\mathrm{e}7 \pm 2.88\mathrm{e}6$ | |
| | MARTA-VDN | $5.58 \pm 0.54$ | +9.8 | $1.51\mathrm{e}7 \pm 1.77\mathrm{e}6$ | +13.5 |
| SMAC-2s3z | QMIX | $4.87 \pm 0.032$ | | $1.39\mathrm{e}7 \pm 7.03\mathrm{e}6$ | |
| | MARTA-QMIX | $6.08 \pm 0.638$ | +24.8 | $1.59\mathrm{e}7 \pm 6.27\mathrm{e}6$ | +2.3 |
| | VDN | $6.05 \pm 0.35$ | | $1.76\mathrm{e}7 \pm 6.36\mathrm{e}6$ | |
| | MARTA-VDN | $7.31 \pm 0.57$ | +20.8 | $1.82\mathrm{e}7 \pm 9.22\mathrm{e}6$ | +3.4 |

# C. Algorithms

In this section, we provide the pseudocode for 3 variants of MARTA, namely an actor-critic variant of MARTA called MARTA-AC (Algorithm 1) a Q learning variant called MARTA-Q (Algorithm 2) and, lastly a version of MARTA-AC that accommodates budget constraints called MARTA-Budget (Algorithm 3).

---

**Algorithm 1** **M**ulti-**A**gent **R**obust **T**raining **A**lgorithm (MARTA)-AC

---

1: **Input:** Stepsize $\alpha$, batch size $B$, episodes $K$, steps per episode $T$, mini-epochs $e$, malfunction cost $c$, Termination probability parameter $p$ (Bernoulli distributed).

2: **Initialise:** Policy network (acting) $\pi$, Policy network (switching) $\mathfrak{g}$,
   Policy network (adversary) $(\sigma^1, \ldots, \sigma^N) = \boldsymbol{\sigma}$, Critic network (acting )$V_{\boldsymbol{\pi}}$, Critic network (switching )$V_{\mathfrak{g}}$, Critic network (adversary )$V_{\boldsymbol{\sigma}}$, for any $t < 0$ set termination probability $p_t \equiv 1$.

3: Given reward objective function, $\mathcal{R}$, initialise Rollout Buffers $\mathcal{B}_{\pi}, \mathcal{B}_{\mathfrak{g}}$ (use Replay Buffer for SAC), $\mathcal{B}_{\boldsymbol{\sigma}}$.

4: **for** $N_{episodes}$ **do**

5:     Reset state $s_0$, Reset Rollout Buffers $\mathcal{B}_{\pi}, \mathcal{B}_{\mathfrak{g}}, \mathcal{B}_{\boldsymbol{\sigma}}$

6:     **for** $t = 0, 1, \ldots$ **do**

7:         Sample $(f_t^1, \ldots, f_t^N) = \boldsymbol{f}_t \sim \boldsymbol{\sigma}(\cdot|s_t)$, $\boldsymbol{a}_t \sim \boldsymbol{\pi}(\cdot|s_t)$, $g_t \sim \mathfrak{g}(\cdot|s_t)$, $p_t \sim \mathrm{Bern}(p)$

8:         **if** $(1 - p_{t-1})g_{t-1} > 0$ (hence $(1 - p_{t-1})g_{t-1} = i \in \mathcal{N}$) **then**

9:             Set $g_t \equiv g_{t-1}$. Apply $f_t^i$ and $a_t^{-i}$ where $i \equiv g_{t-1}$ so $s_{t+1} \sim P(\cdot|(f_t^i, a_t^{-i}), s_t)$,

10:             Receive rewards $\boldsymbol{r}_{S,t} = -c - \boldsymbol{r}_t$ and $\boldsymbol{r}_t$ where $\boldsymbol{r}_t \sim \mathcal{R}(s_t, (f_t^i, a_t^{-i}))$

11:             Store $(s_t, (f_t^i, a_t^{-i}), s_{t+1}, \boldsymbol{r}_t)$, $(s_t, (f_t^i, a_t^{-i}), s_{t+1}, -\boldsymbol{r}_t)$ and $(s_t, g_t, s_{t+1}, \boldsymbol{r}_{S,t})$ in $\mathcal{B}_{\boldsymbol{\pi}}, \mathcal{B}_{\boldsymbol{\sigma}}$ and $\mathcal{B}_{\mathfrak{g}}$ respectively.

12:         **else**

13:             **if** $g_t = i \in \{1, \ldots, N\}$ **then**

14:                 Apply $f_t^i$ and $a_t^{-i}$ so $s_{t+1} \sim P(\cdot|(f_t^i, a_t^{-i}), s_t)$,

15:                 Receive rewards $\boldsymbol{r}_{S,t} = -c - \boldsymbol{r}_t$ and $\boldsymbol{r}_t$ where $\boldsymbol{r}_t \sim \mathcal{R}(s_t, (f_t^i, a_t^{-i}))$

16:                 Store $(s_t, (f_t^i, a_t^{-i}), s_{t+1}, \boldsymbol{r}_t)$, $(s_t, (f_t^i, a_t^{-i}), s_{t+1}, -\boldsymbol{r}_t)$ and $(s_t, g_t, s_{t+1}, \boldsymbol{r}_{S,t})$ in $\mathcal{B}_{\boldsymbol{\pi}}, \mathcal{B}_{\boldsymbol{\sigma}}$ and $\mathcal{B}_{\mathfrak{g}}$ respectively.

17:             **else**

18:                 Apply the actions $\boldsymbol{a}_t \sim \boldsymbol{\pi}(\cdot|s_t)$ so $s_{t+1} \sim P(\cdot|\boldsymbol{a}_t, s_t)$,

19:                 Receive rewards $\boldsymbol{r}_{S,t} = -\boldsymbol{r}_t$ and $\boldsymbol{r}_t$ where $\boldsymbol{r}_t \sim \mathcal{R}(s_t, \boldsymbol{a}_t)$.

20:             **end if**

21:         **end if**

22:         Store $(s_t, \boldsymbol{a}_t, s_{t+1}, \boldsymbol{r}_t)$ and $(s_t, g_t, s_{t+1}, \boldsymbol{r}_{S,t})$ in $\mathcal{B}_{\boldsymbol{\pi}}$ and $\mathcal{B}_{\mathfrak{g}}$ respectively.

23:     **end for**

24:     **// Learn the individual policies**

25:     Update policy $\pi$ and critic $V_{\boldsymbol{\pi}}$ networks using $\mathfrak{B}_{\boldsymbol{\pi}}$

26:     Update policy $\mathfrak{g}$ and critic $V_{\mathfrak{g}}$ networks using $\mathfrak{B}_{\mathfrak{g}}$

27:     Update policy $\boldsymbol{\sigma}$ and critic $V_{\boldsymbol{\sigma}}$ networks using $\mathfrak{B}_{\boldsymbol{\sigma}}$

28: **end for**

---

---

**Algorithm 2** **M**ulti-**A**gent **R**obust **T**raining **A**lgorithm (MARTA)- Q

---

1: **Input:** Constant $\epsilon \geq 0$,
2: **Initialise:** Q-function, $\boldsymbol{Q}_0$, set termination probability $p_t \equiv 1$ for any $t < 0$
3: $n \leftarrow 0$
4: **for** $t = 0, 1, \ldots$ **do**
5:     Estimate $\boldsymbol{a}_t \in \arg\max \boldsymbol{Q}_n(s_t, \boldsymbol{a}_t)$
6:     Estimate $f_t^i \in \underset{i \in \mathcal{N}, a_t^i \in \mathcal{A}^i}{\arg\min} \underset{a_t^{-i} \in \mathcal{A}^{-i}}{\max} Q_S(s, (a_t^i, a_t^{-i}), g)$
7:     **if** $(1 - p_{t-1})g_{t-1} > 0$ (i.e. $(1 - p_{t-1})g_{t-1} \in \mathcal{N}$) **then**
8:         Set $g_t \equiv g_{t-1}$. Apply $f_t^i$ and $a_t^{-i}$ where $i \equiv g_{t-1}$ so $s_{t+1} \sim P(\cdot|(f_t^i, a_t^{-i}), s_t)$,
9:     **else**
10:         **if** $\hat{\mathcal{M}}Q_n \geq \boldsymbol{Q}_n$ **then**
11:             Apply $f_t^i$ and $a_t^{-i}$ so $s_{t+1} \sim P(\cdot|(f_t^i, a_t^{-i}), s_t)$
12:             Receive rewards $\boldsymbol{r}_{S,t} = -c - \boldsymbol{r}_t$ and $\boldsymbol{r}_t$ where $\boldsymbol{r}_t \sim \mathcal{R}(s_t, (f_t^i, a_t^{-i}))$
13:         **else**
14:             Apply the actions $\boldsymbol{a}_t$ so $s_{t+1} \sim P(\cdot|\boldsymbol{a}_t, s_t)$,
15:             Receive rewards $\boldsymbol{r}_{S,t} = -\boldsymbol{r}_t$ and $\boldsymbol{r}_t$ where $\boldsymbol{r}_t \sim \mathcal{R}(s_t, \boldsymbol{a}_t)$.
16:         **end if**
17:     **end if**
18: **end for**
19: **// Learn $\hat{Q}$**
20: Update $\boldsymbol{Q}_n$ function according to the update rule equation 2

---

**Algorithm 3** **M**ulti-**A**gent **R**obust **T**raining **A**lgorithm (MARTA)-Budget

---

1: **Input:** Stepsize $\alpha$, batch size $B$, episodes $K$, steps per episode $T$, mini-epochs $e$, malfunction cost $c$.
2: **Initialise:** Policy network (acting) $\boldsymbol{\pi}$, Policy network (switching) $\mathfrak{g}$,
    Policy network (adversary) $(\sigma^1, \ldots, \sigma^N) = \boldsymbol{\sigma}$, Critic network (acting )$V_{\boldsymbol{\pi}}$, Critic network (switching )$V_{\mathfrak{g}}$, Critic
    network (adversary )$V_{\boldsymbol{\sigma}}$.
3: Given reward objective function, $\mathcal{R}$, initialise Rollout Buffers $\mathcal{B}_{\boldsymbol{\pi}}, \mathcal{B}_{\mathfrak{g}}$ (use Replay Buffer for SAC), $\mathcal{B}_{\boldsymbol{\sigma}}$.
4: **for** $N_{episodes}$ **do**
5:     Reset state $s_0$, Reset Rollout Buffers $\mathcal{B}_{\boldsymbol{\pi}}, \mathcal{B}_{\mathfrak{g}}, \mathcal{B}_{\boldsymbol{\sigma}}$
6:     **for** $t = 0, 1, \ldots$ **do**
7:         Sample $(f_t^1, \ldots, f_t^N) = \boldsymbol{f}_t \sim \boldsymbol{\sigma}(\cdot|\boldsymbol{s}_t), \boldsymbol{a}_t \sim \boldsymbol{\pi}(\cdot|\boldsymbol{s}_t), g_t \sim \mathfrak{g}(\cdot|\boldsymbol{s}_t, \boldsymbol{f}_t, \boldsymbol{a}_t)\ \boldsymbol{s}_t = (s_t, y_t)$
8:         **if** $g_t = i \in \{1, \ldots, N\}$ **then**
9:             Apply $f_t^i$ and $a_t^{-i}$ so $s_{t+1} \sim \widetilde{P}(\cdot|(f_t^i, a_t^{-i}), \boldsymbol{s}_t)$
10:            Receive rewards $\widetilde{\boldsymbol{r}}_{S,t} = -c - \boldsymbol{r}_t$ and $\widetilde{\boldsymbol{r}}_t$ where $\widetilde{\boldsymbol{r}}_t \sim \widetilde{\mathcal{R}}(\boldsymbol{s}_t, (f_t^i, a_t^{-i}))$,
11:            Store $(\boldsymbol{z}_t, f_t^i, a_t^{-i}, \boldsymbol{s}_{t+1}, \boldsymbol{r}_{S,t}, \boldsymbol{r}_t)$ in $\mathcal{B}_{\boldsymbol{\pi}}, \mathcal{B}_{\boldsymbol{\sigma}}, \mathcal{B}_{\mathfrak{g}}$ where $\boldsymbol{z}_t := (\boldsymbol{s}_t, g_t), \ \boldsymbol{s}_t = (s_t, y_t)$.
12:         **else**
13:            Apply the actions $\boldsymbol{a}_t \sim \boldsymbol{\pi}(\cdot|\boldsymbol{s}_t)$ so $s_{t+1} \sim P(\cdot|\boldsymbol{a}_t, \boldsymbol{s}_t)$,
14:            Receive rewards $\widetilde{\boldsymbol{r}}_{S,t} = -\widetilde{\boldsymbol{r}}_t$ and $\widetilde{\boldsymbol{r}}_t$ where $\widetilde{\boldsymbol{r}}_t \sim \widetilde{\mathcal{R}}(\boldsymbol{s}_t, \boldsymbol{a}_t)$.
15:         **end if**
16:         Store $(\boldsymbol{z}_t, \boldsymbol{a}_t, \boldsymbol{s}_{t+1}, \boldsymbol{r}_{S,t}, \boldsymbol{r}_t)$ in $\mathcal{B}_{\boldsymbol{\pi}}, \mathcal{B}_{\boldsymbol{\sigma}}, \mathcal{B}_{\mathfrak{g}}$.
17:     **end for**
18:     **// Learn the individual policies**
19:     Update policy $\boldsymbol{\pi}$ and critic $V_{\boldsymbol{\pi}}$ networks using $\mathfrak{B}_{\boldsymbol{\pi}}$
20:     Update policy $\mathfrak{g}$ and critic $V_{\mathfrak{g}}$ networks using $\mathfrak{B}_{\mathfrak{g}}$
21:     Update policy $\boldsymbol{\sigma}$ and critic $V_{\boldsymbol{\sigma}}$ networks using $\mathfrak{B}_{\boldsymbol{\sigma}}$
22: **end for**

---

### C.1. Computational Requirements

Most experiments were conducted on shared compute clusters using NVIDIA A40 GPUs with 46GB of memory. MARTA uses the same backbone architectures and hyperparameters as the corresponding MARL baselines, with an additional lightweight Switcher–Adversary module during training. This adds only a modest training-time overhead and does not introduce any additional inference-time cost.

### C.2. Hyperparameter Settings

In the table below we report all hyperparameters used in our experiments. Hyperparameter values in square brackets indicate ranges of values that were used for performance tuning.

| | |
|---|---|
| Clip Gradient Norm | 1 |
| $\gamma_E$ | 0.99 |
| $\lambda$ | 0.95 |
| Learning rate | $1\text{x}10^{-4}$ |
| Number of minibatches | 4 |
| Number of optimisation epochs | 4 |
| Number of parallel actors | 16 |
| Optimisation algorithm | Adam |
| Rollout length | 128 |
| Sticky action probability | 0.25 |
| Use Generalised Advantage Estimation | True |
| Coefficient of extrinsic reward | [1, 5] |
| Coefficient of intrinsic reward | [1, 2, 5, 10, 20, 50] |
| Switcher discount factor | 0.99 |
| Probability of terminating option | [0.5, 0.75, 0.8, 0.9, 0.95] |
| $L$ function output size | [2, 4, 8, 16, 32, 64, 128, 256] |

## D. Additional Theoretical Results

**Theorem D.1.** *Let $v : \mathcal{S} \to \mathbb{R}$ then the sequence of Bellman operators acting on $v$ converges to the solution of the game, that is to say for any $s \in \mathcal{S}$ the following holds:*

$$\lim_{k \to \infty} T^k v(s | \boldsymbol{\pi}, \mathfrak{g}) = v^*(s), \tag{2}$$

*where* $v^*(s) = \min_{\hat{\mathfrak{g}}} \max_{\hat{\boldsymbol{\pi}} \in \Pi} v(s | \hat{\boldsymbol{\pi}}, \hat{\mathfrak{g}}) = \max_{\hat{\boldsymbol{\pi}} \in \Pi} \min_{\hat{\mathfrak{g}}} v(s | \hat{\boldsymbol{\pi}}, \hat{\mathfrak{g}})$, *the operator* $T$ *is given by* $Tv(s | \boldsymbol{\pi}, \mathfrak{g}) = \min \left\{ \hat{\mathcal{M}} Q_S, \max_{\boldsymbol{a} \in \mathcal{A}} \left[ \mathcal{R}(s, \boldsymbol{a}) + \gamma \sum_{s' \in \mathcal{S}} P(s'; \boldsymbol{a}, s) v(s' | \boldsymbol{\pi}, \mathfrak{g}) \right] \right\}$ *where* $\hat{\mathcal{M}} Q_S(s, \boldsymbol{a}, g | \boldsymbol{\pi}, \mathfrak{g}) := c + \min_{i \in \mathcal{N}, a^i \in \mathcal{A}^i} \max_{a^{-i} \in \mathcal{A}^{-i}} Q_S(s, (a^i, a^{-i}), g | \boldsymbol{\pi}, \mathfrak{g})$.

We introduce the associated reward functions $\widetilde{\mathcal{R}} : \mathcal{X} \times \mathcal{A} \to \mathcal{P}(D)$ and $\widetilde{\mathcal{R}}_S : \mathcal{X} \times \mathcal{A} \times \mathcal{N} \times \{0\} \to \mathcal{P}(D)$ and the probability transition function $\widetilde{P} : \mathcal{X} \times \mathcal{A} \times \mathcal{X} \to [0, 1]$ whose state space input is now replaced by $\mathcal{X}$ and Switcher value function for the budgeted game $\widetilde{\mathcal{G}}$. We now prove MARTA-B generates best-response FT policies with a fault-activation budget.

**Theorem D.2.** *Consider the budgeted problem $\widetilde{\mathcal{G}}$, then for any $\widetilde{v} : \mathcal{X} \to \mathbb{R}$, the solution of $\widetilde{\mathcal{G}}$ is given by*

$$\lim_{k \to \infty} \tilde{T}_G^k \widetilde{v}_S = \min_{\hat{\mathfrak{g}}} \max_{\hat{\boldsymbol{\pi}}} \widetilde{v}_S(\cdot | \hat{\boldsymbol{\pi}}, \hat{\mathfrak{g}}) = \max_{\hat{\boldsymbol{\pi}}} \min_{\hat{\mathfrak{g}}} \widetilde{v}_S(\cdot | \hat{\boldsymbol{\pi}}, \hat{\mathfrak{g}}),$$

*where Switcher's optimal policy takes the Markovian form $\widetilde{\mathfrak{g}}(\cdot | \boldsymbol{x})$ for any $\boldsymbol{y} \equiv (y, s) \in \mathcal{X}$.*

Theorem D.2 proves, under standard assumptions, MARTA converges to the solution of Switcher's problem and the dec-POMDP under a Switcher fault-activation budget constraint.

# E. Proof of Technical Results

## E.1. Notation & Assumptions

We assume that $\mathcal{S}$ is defined on a probability space $(\Omega, \mathcal{F}, \mathbb{P})$ and any $s \in \mathcal{S}$ is measurable with respect to the Borel $\sigma$-algebra associated with $\mathbb{R}^p$. We denote the $\sigma$-algebra of events generated by $\{s_t\}_{t \geq 0}$ by $\mathcal{F}_t \subset \mathcal{F}$. In what follows, we denote by $(\mathcal{Y}, \|\|)$ any finite normed vector space and by $\mathcal{H}$ the set of all measurable functions. The results of the paper are built under the following assumptions which are standard within RL and stochastic approximation methods:

**Assumption 1.** The stochastic process governing the system dynamics is ergodic, that is the process is stationary and every invariant random variable of $\{s_t\}_{t \geq 0}$ is equal to a constant with probability $1$.

**Assumption 2**. The agents' reward function $\mathcal{R}$ is in $L_2$.

**Assumption 3.** For any Switcher policy $\mathfrak{g}$, the total number of interventions is $K < \infty$.

We begin the analysis with some preliminary results and definitions required for proving our main results.

**Definition E.1.** A.1 Given a norm $\| \cdot \|$, an operator $T : \mathcal{Y} \to \mathcal{Y}$ is a contraction if there exists some constant $c \in [0, 1[$ for which for any $J_1, J_2 \in \mathcal{Y}$ the following bound holds: $\|TJ_1 - TJ_2\| \leq c\|J_1 - J_2\|$.

**Definition E.2.** A.2 An operator $T : \mathcal{Y} \to \mathcal{Y}$ is non-expansive if $\forall J_1, J_2 \in \mathcal{Y}$ the following bound holds: $\|TJ_1 - TJ_2\| \leq \|J_1 - J_2\|$.

**Lemma E.3.** *(Mguni, 2019) For any $f : \mathcal{Y} \to \mathbb{R} : \mathcal{Y} \to \mathbb{R}$, we have that the following inequality holds:*

$$\left\| \max_{a \in \mathcal{Y}} f(a) - \max_{a \in \mathcal{Y}} g(a) \right\| \leq \max_{a \in \mathcal{Y}} \|f(a) - g(a)\|. \tag{3}$$

**Lemma E.4.** *A.4(Tsitsiklis & Van Roy, 1999) The probability transition kernel $P$ is non-expansive so that if $\forall J_1, J_2 \in \mathcal{Y}$ the following holds: $\|PJ_1 - PJ_2\| \leq \|J_1 - J_2\|$.*

**Lemma E.5.** *A.1 Define $\mathrm{val}^+[f] := \min_{b \in \mathbb{B}} \max_{a \in \mathbb{A}} f(a, b)$ and define $\mathrm{val}^-[f] := \max_{a \in \mathbb{A}} \min_{b \in \mathbb{B}} f(a, b)$, then for any $b \in \mathbb{B}$ we have that for any $f, g \in \mathbb{L}$ and for any $k \in \mathbb{R}_{>0}$:*

$$\left| \max_{a \in \mathbb{A}} f(a, b) - \max_{a \in \mathbb{A}} g(a, b) \right| \leq k \implies \left| \mathrm{val}^-[f] - \mathrm{val}^-[g] \right| \leq k.$$

**Lemma E.6.** *A.2 For any $f, g, h \in \mathbb{L}$ and for any $k \in \mathbb{R}_{>0}$ we have that:*

$$\|f - g\| \leq k \implies \|\min\{f, h\} - \min\{g, h\}\| \leq k.$$

**Lemma E.7.** *A.3 Let the functions $f, g, h \in \mathbb{L}$ then*

$$\|\max\{f, h\} - \max\{g, h\}\| \leq \|f - g\|. \tag{4}$$

*Proof of Lemma E.5.* We begin by noting the following inequality for any $f : \mathcal{V} \times \mathcal{V} \to \mathbb{R}, g : \mathcal{V} \times \mathcal{V} \to \mathbb{R}$ s.th. $f, g \in \mathbb{L}$ we have by Lemma 5, that for all $b \in \mathcal{V}$:

$$\left| \max_{a \in \mathcal{V}} f(a, b) - \max_{a \in \mathcal{V}} g(a, b) \right| \leq \max_{a \in \mathcal{V}} |f(a, b) - g(a, b)|. \tag{5}$$

From (5) we can straightforwardly derive the fact that for any $b \in \mathcal{V}$:

$$\left| \min_{a \in \mathcal{V}} f(a, b) - \min_{a \in \mathcal{V}} g(a, b) \right| \leq \max_{a \in \mathcal{V}} |f(a, b) - g(a, b)|, \tag{6}$$

(this can be seen by negating each of the functions in (5) and using the properties of the $\max$ operator).

Assume that for any $b \in \mathcal{V}$ the following inequality holds:

$$\max_{a \in \mathcal{V}} |f(a, b) - g(a, b)| \leq k \tag{7}$$

Since (6) holds for any $b \in \mathcal{V}$ and, by (5), we have in particular that

$$\left| \max_{b \in \mathcal{V}} \min_{a \in \mathcal{V}} f(a,b) - \max_{b \in \mathcal{V}} \min_{a \in \mathcal{V}} g(a,b) \right|$$

$$\leq \max_{b \in \mathcal{V}} \left| \min_{a \in \mathcal{V}} f(a,b) - \min_{a \in \mathcal{V}} g(a,b) \right|$$

$$\leq \max_{b \in \mathcal{V}} \max_{a \in \mathcal{V}} |f(a,b) - g(a,b)| \leq k, \tag{8}$$

whenever (7) holds which gives the required result. $\qquad \square$

The following result fully characterises Switcher's policy $\mathfrak{g}$ in terms of an obstacle condition which can be determined at each state:

**Proposition E.8.** *Denote by* $\boldsymbol{a} \equiv (a^i, a^{-i}) \in \boldsymbol{\mathcal{A}}$, *the optimal Switcher policy* $\mathfrak{g}^*$ *is given by* $\mathfrak{g}^*(\cdot|s) = \arg\min_{i \in \mathcal{N}} \left[ \min_{a^i \in \mathcal{A}^i} \max_{a^{-i} \in \mathcal{A}^{-i}} Q_S(s, \boldsymbol{a}, g|\cdot) \right] \mathbf{1}\left( \max_{\boldsymbol{a} \in \boldsymbol{\mathcal{A}}} Q_S(s, \boldsymbol{a}, g|\cdot) - \hat{\mathcal{M}} Q_S(s, \boldsymbol{a}, g|\cdot) \right)$ *for any* $s \in \mathcal{S}$ *where* $\mathbf{1}_{\mathbb{R}_+}(x) = 1$ *if* $x > 0$ *and* $0$ *otherwise.*

Prop. E.8 provides characterisation of where Switcher should activate Adversary and which agent $i \in \mathcal{N}$ should be activated. The condition allows for the characterisation to be evaluated online during the learning phase.

## Proof of Theorem D.1

*Proof.* We begin by recalling the definition of the intervention operator $\hat{\mathcal{M}}$ for any $s \in \mathcal{S}$:

$$\hat{\mathcal{M}} Q_S(s, \boldsymbol{a}, g|\cdot) := c + \min_{i \in \mathcal{N}, a^i \in \mathcal{A}^i} \max_{a^{-i} \in \mathcal{A}^{-i}} Q_S(s, (a^i, a^{-i}), g|\cdot) \tag{9}$$

Secondly, recall that the Bellman operator for the game $\mathcal{G}$ acting on a function $v_S : \mathcal{S} \to \mathbb{R}$ is:

$$T v_S(s) := \min \left\{ \hat{\mathcal{M}} Q_S, \max_{\boldsymbol{a} \in \boldsymbol{\mathcal{A}}} \left[ \mathcal{R}_S + \gamma \sum_{s' \in \mathcal{S}} P(s'; \cdot) v_S(s') \right] \right\} \tag{10}$$

It suffices to prove that $T$ is a contraction operator. Thereafter, we use both results to prove the existence of a fixed point for $\mathcal{G}$ as a limit point of a sequence generated by successively applying the Bellman operator to a test value function. Therefore our next result shows that the following bounds holds:

**Lemma E.9.** *The Bellman operator $T$ is a contraction so that for any real-valued maps $v_S, v'_S$, the following bound holds:* $\|T v_S - T v'_S\| \leq \gamma \|v_S - v'_S\|$.

In the following proofs, we use the shorthand notation: $\mathcal{P}^{\boldsymbol{a}}_{ss'} =: \sum_{s' \in \mathcal{S}} P(s'; \boldsymbol{a}, s)$ and $\mathcal{P}^{\boldsymbol{\pi}}_{ss'} =: \sum_{\boldsymbol{a} \in \boldsymbol{\mathcal{A}}} \boldsymbol{\pi}(\boldsymbol{a}|s) \mathcal{P}^{\boldsymbol{a}}_{ss'}$. To prove that $T$ is a contraction, we consider the three cases produced by equation 10, that is to say we prove the following statements:

i) $\qquad \left| \max_{\boldsymbol{a} \in \boldsymbol{\mathcal{A}}} \left( \mathcal{R}_S(s_t, \boldsymbol{a}, g) + \gamma \mathcal{P}^{\boldsymbol{a}}_{s' s_t} v_S(s') \right) - \max_{\boldsymbol{a} \in \boldsymbol{\mathcal{A}}} \left( \mathcal{R}_S(s_t, \boldsymbol{a}, g) + \gamma \mathcal{P}^{\boldsymbol{a}}_{s' s_t} v'_S(s') \right) \right| \leq \gamma \|v_S - v'_S\|$

ii) $\qquad \left\| \hat{\mathcal{M}} Q_S - \hat{\mathcal{M}} Q'_S \right\| \leq \gamma \|v_S - v'_S\|, \qquad\qquad .$

iii) $\qquad \left\| \hat{\mathcal{M}} Q_S - \max_{\boldsymbol{a} \in \boldsymbol{\mathcal{A}}} [\mathcal{R}_S(s_t, \boldsymbol{a}, g) + \gamma \mathcal{P}^{\boldsymbol{a}}_{s' s} v'_S] \right\| \leq \gamma \|v_S - v'_S\|$.

We begin by proving i).

Indeed, for any $\boldsymbol{a} \in \mathcal{A}$ and $\forall s_t \in \mathcal{S}, \forall s' \in \mathcal{S}$ we have that

$$\left| \max_{\boldsymbol{a} \in \mathcal{A}} \left( \mathcal{R}_S(s_t, \boldsymbol{a}, g) + \gamma \mathcal{P}^\pi_{s' s_t} v_S(s') \right) - \max_{\boldsymbol{a} \in \mathcal{A}} \left( \mathcal{R}_S(s_t, \boldsymbol{a}, g) + \gamma \mathcal{P}^{\boldsymbol{a}}_{s' s_t} v'_S(s') \right) \right|$$

$$\leq \max_{\boldsymbol{a} \in \mathcal{A}} \left| \gamma \mathcal{P}^{\boldsymbol{a}}_{s' s_t} v_S(s') - \gamma \mathcal{P}^{\boldsymbol{a}}_{s' s_t} v'_S(s') \right|$$

$$\leq \gamma \left\| P v_S - P v'_S \right\|$$

$$\leq \gamma \left\| v_S - v'_S \right\|,$$

using the non-expansiveness of the operator $P$ and Lemma E.3.

We now prove ii). Using the definition of $\mathcal{M}$ we have that for any $s \in \mathcal{S}$

$$\left| (\hat{\mathcal{M}} Q_S - \hat{\mathcal{M}} Q'_S)(s, \boldsymbol{a}) \right|$$

$$= \left| \min_{i \in \mathcal{N}, a^i \in \mathcal{A}^i} \max_{a^{-i} \in \mathcal{A}^{-i}} \left( \mathcal{R}_S(s, \boldsymbol{a}, g) + c + \gamma \mathcal{P}^{\boldsymbol{a}}_{s' s} v_S(s) \right) - \min_{i \in \mathcal{N}, a^i \in \mathcal{A}^i} \max_{a^{-i} \in \mathcal{A}^{-i}} \left( \mathcal{R}_S(s, \boldsymbol{a}, g) + c + \gamma \mathcal{P}^{\boldsymbol{a}}_{s' s} v'_S(s) \right) \right|$$

$$\leq \gamma \max_{i \in \mathcal{N}, a^i \in \mathcal{A}^i} \max_{a^{-i} \in \mathcal{A}^{-i}} \left| \gamma \mathcal{P}^{a^i} \mathcal{P}^{a^{-i}} v_S(s) - \mathcal{P}^{a^i} \mathcal{P}^{a^{-i}} v'_S(s) \right|$$

$$\leq \gamma \max_{\boldsymbol{a} \in \mathcal{A}} \left| \mathcal{P}^{a^i} \mathcal{P}^{a^{-i}} v_S(s) - \mathcal{P}^{a^i} \mathcal{P}^{a^{-i}} v'_S(s) \right|$$

$$\leq \gamma \left\| P v_S - P v'_S \right\|$$

$$\leq \gamma \left\| v_S - v'_S \right\|,$$

using equation 6 in the second step and the fact that $P$ is non-expansive.

We now prove iii). In the proof of iii), we use the following equivalent representation of Switcher objective which is to find $(\hat{\boldsymbol{\pi}}, \hat{\mathfrak{g}})$ such that

$$(\hat{\boldsymbol{\pi}}, \hat{\mathfrak{g}}) \in \arg \max_{\boldsymbol{\pi}, \mathfrak{g}} \mathfrak{v}_S(s | \boldsymbol{\pi}, \mathfrak{g})$$

$$\mathfrak{v}_S(s | \boldsymbol{\pi}, \mathfrak{g}) = \mathbb{E}_{g \sim \mathfrak{g}} \left[ \sum_{t=0}^{\infty} \gamma^t \left( -\mathcal{R}(s_t, \boldsymbol{a}_t) - c \cdot \mathbf{1}_{\mathcal{N}}(g(s_t)) \right) \Big| s_0 = s; \boldsymbol{a} \sim \boldsymbol{\pi} \right].$$

Now the intervention operator transforms as $\hat{M}$:

$$\hat{M} Q_S(s, \boldsymbol{a}, g | \cdot) := -c + \max_{i \in \mathcal{N}, a^i \in \mathcal{A}^i} \min_{a^{-i} \in \mathcal{A}^{-i}} Q_S(s, (a^i, a^{-i}), g | \cdot), \tag{11}$$

for any $s \in \mathcal{S}$ and any $\boldsymbol{a} \in \mathcal{A}$. Similarly, the Bellman operator acting on a function $v_S : \mathcal{S} \to \mathbb{R}$ becomes:

$$\mathcal{T} v_S(s) := \min \left\{ \hat{M} Q_S, \min_{\boldsymbol{a} \in \mathcal{A}} \left[ -\mathcal{R}_S + \gamma \sum_{s' \in \mathcal{S}} P(s'; \cdot) v_S(s') \right] \right\}, \forall s \in \mathcal{S}. \tag{12}$$

Therefore, to prove (iii), we must show that:

$$\left\| \hat{M} Q_S - \min_{\boldsymbol{a} \in \mathcal{A}} \left[ -\mathcal{R}_S(s, \boldsymbol{a}, g) + \gamma \mathcal{P}^{\boldsymbol{a}}_{s' s} v'_S \right] \right\| \leq \gamma \left\| v_S - v'_S \right\|. \tag{13}$$

We split the proof of the statement into two cases:

**Case 1:** First, assume that for any $s \in \mathcal{S}$ and $\forall \boldsymbol{a} \in \mathcal{A}$ the following inequality holds:

$$\hat{M} Q_S(s, \boldsymbol{a}, g | \cdot) - \min_{\boldsymbol{a} \in \mathcal{A}} \left( -\mathcal{R}_S(s, \boldsymbol{a}, g) + \gamma \mathcal{P}^{\boldsymbol{a}}_{s' s} v'_S(s') \right) < 0. \tag{14}$$

We now observe the following:

$$\hat{M}Q_S(s,\boldsymbol{a},g|\cdot) - \min_{\boldsymbol{a}\in\mathcal{A}} \left(-\mathcal{R}_S(s,\boldsymbol{a},g) + \gamma\mathcal{P}^{\boldsymbol{a}}_{s's}v'_S(s')\right)$$

$$\leq \max\left\{\min_{\boldsymbol{a}\in\mathcal{A}} \left(-\mathcal{R}_S(s,\boldsymbol{a},g) + \gamma\mathcal{P}^{\boldsymbol{a}}_{s's}v_S(s')\right), \hat{M}Q_S(s,\boldsymbol{a},g|\cdot)\right\} - \min_{\boldsymbol{a}\in\mathcal{A}} \left(-\mathcal{R}_S(s,\boldsymbol{a},g) + \gamma\mathcal{P}^{\boldsymbol{a}}_{s's}v'_S(s')\right)$$

$$\leq \left| \max\left\{\min_{\boldsymbol{a}\in\mathcal{A}} \left(-\mathcal{R}_S(s,\boldsymbol{a},g) + \gamma\mathcal{P}^{\boldsymbol{a}}_{s's}v_S(s')\right), \hat{M}Q_S(s,\boldsymbol{a},g|\cdot)\right\} \right.$$

$$- \max\left\{\min_{\boldsymbol{a}\in\mathcal{A}} \left(-\mathcal{R}_S(s,\boldsymbol{a},g) + \gamma\mathcal{P}^{\boldsymbol{a}}_{s's}v'_S(s')\right), \hat{M}Q_S(s,\boldsymbol{a},g|\cdot)\right\}$$

$$\left. + \max\left\{\min_{\boldsymbol{a}\in\mathcal{A}} \left(-\mathcal{R}_S(s,\boldsymbol{a},g) + \gamma\mathcal{P}^{\boldsymbol{a}}_{s's}v'_S(s')\right), \hat{M}Q_S(s,\boldsymbol{a},g|\cdot)\right\} - \min_{\boldsymbol{a}\in\mathcal{A}} \left(-\mathcal{R}_S(s,\boldsymbol{a},g) + \gamma\mathcal{P}^{\boldsymbol{a}}_{s's}v'_S(s')\right) \right|$$

$$\leq \left| \max\left\{\min_{\boldsymbol{a}\in\mathcal{A}} \left(-\mathcal{R}_S(s,\boldsymbol{a},g) + \gamma\mathcal{P}^{\boldsymbol{a}}_{s's}v_S(s')\right), \hat{M}Q_S(s,\boldsymbol{a},g|\cdot)\right\} \right.$$

$$\left. - \max\left\{\min_{\boldsymbol{a}\in\mathcal{A}} \left(-\mathcal{R}_S(s,\boldsymbol{a},g) + \gamma\mathcal{P}^{\boldsymbol{a}}_{s's}v'_S(s')\right), \hat{M}Q_S(s,\boldsymbol{a},g|\cdot)\right\} \right|$$

$$+ \left| \max\left\{\min_{\boldsymbol{a}\in\mathcal{A}} \left(-\mathcal{R}_S(s,\boldsymbol{a},g) + \gamma\mathcal{P}^{\boldsymbol{a}}_{s's}v'_S(s')\right), \hat{M}Q_S(s,\boldsymbol{a},g|\cdot)\right\} - \min_{\boldsymbol{a}\in\mathcal{A}} \left(-\mathcal{R}_S(s,\boldsymbol{a},g) + \gamma\mathcal{P}^{\boldsymbol{a}}_{s's}v'_S(s')\right) \right|$$

$$\leq \gamma\max_{\boldsymbol{a}\in\mathcal{A}} |\mathcal{P}^{\boldsymbol{a}}_{s's}v_S(s') - \mathcal{P}^{\boldsymbol{a}}_{s's}v'_S(s')| + \left| \max\left\{0, \hat{M}Q_S(s,\boldsymbol{a},g|\cdot) - \min_{\boldsymbol{a}\in\mathcal{A}} \left(-\mathcal{R}_S(s,\boldsymbol{a},g) + \gamma\mathcal{P}^{\boldsymbol{a}}_{s's}v'_S(s')\right)\right\} \right|$$

$$\leq \gamma \|Pv_S - Pv'_S\|$$

$$\leq \gamma \|v_S - v'_S\|,$$

where we have used the fact that for any scalars $a,b,c$ we have that $|\max\{a,b\} - \max\{b,c\}| \leq |a-c|$, the non-expansiveness of $P$ and Lemma 5.

**Case 2:** Let us now consider the case:

$$\hat{M}Q_S(s,\boldsymbol{a},g|\cdot) - \min_{\boldsymbol{a}\in\mathcal{A}} \left(-\mathcal{R}_S(s,\boldsymbol{a},g) + \gamma\mathcal{P}^{\boldsymbol{a}}_{s's}v'_S(s')\right) \geq 0.$$

For this case, first recall that $c > 0$, hence

$$\hat{M}Q_S(s,\boldsymbol{a},g|\cdot) - \min_{\boldsymbol{a}\in\mathcal{A}} \left(-\mathcal{R}_S(s,\boldsymbol{a},g) + \gamma\mathcal{P}^{\boldsymbol{a}}_{s's}v'_S(s')\right)$$

$$\leq \hat{M}Q_S(s,\boldsymbol{a},g|\cdot) - \min_{\boldsymbol{a}\in\mathcal{A}} \left(-\mathcal{R}_S(s,\boldsymbol{a},g) + \gamma\mathcal{P}^{\boldsymbol{a}}_{s's}v'_S(s') - c\right)$$

$$= \max_{i\in\mathcal{N},a^i\in\mathcal{A}^i} \min_{a^{-i}\in\mathcal{A}^{-i}} \left(-\mathcal{R}_S(s,\boldsymbol{a},g) - c + \gamma\mathcal{P}^{a^i}_{s's}\mathcal{P}^{a^{-i}}_{s's}v_S(s')\right) - \min_{\boldsymbol{a}\in\mathcal{A}} \left(-\mathcal{R}_S(s,\boldsymbol{a},g) - c + \gamma\mathcal{P}^{\boldsymbol{a}}_{s's}v'_S(s')\right)$$

$$\leq \max_{i\in\mathcal{N},a^i\in\mathcal{A}^i} \max_{a^{-i}\in\mathcal{A}^{-i}} \left(-\mathcal{R}_S(s,\boldsymbol{a},g) + \gamma\mathcal{P}^{a^i}_{s's}\mathcal{P}^{a^{-i}}_{s's}v_S(s')\right) - \min_{\boldsymbol{a}\in\mathcal{A}} \left(-\mathcal{R}_S(s,\boldsymbol{a},g) + \gamma\mathcal{P}^{\boldsymbol{a}}_{s's}v'_S(s')\right)$$

$$= \max_{i\in\mathcal{N},a^i\in\mathcal{A}^i} \max_{a^{-i}\in\mathcal{A}^{-i}} \left(-\mathcal{R}_S(s,\boldsymbol{a},g) + \gamma\mathcal{P}^{a^i}_{s's}\mathcal{P}^{a^{-i}}_{s's}v_S(s')\right) + \max_{\boldsymbol{a}\in\mathcal{A}} \left(\mathcal{R}_S(s,\boldsymbol{a},g) - \gamma\mathcal{P}^{\boldsymbol{a}}_{s's}v'_S(s')\right)$$

$$\leq \max_{\boldsymbol{a}\in\mathcal{A}} \left(-\mathcal{R}_S(s,\boldsymbol{a},g) + \gamma\mathcal{P}^{\boldsymbol{a}}_{s's}v_S(s')\right) + \max_{\boldsymbol{a}\in\mathcal{A}} \left(\mathcal{R}_S(s,\boldsymbol{a},g) - \gamma\mathcal{P}^{\boldsymbol{a}}_{s's}v'_S(s')\right)$$

$$\leq \gamma\max_{\boldsymbol{a}\in\mathcal{A}} |\mathcal{P}^{\boldsymbol{a}}_{s's}(v_S(s') - v'_S(s'))|$$

$$\leq \gamma |v_S(s') - v'_S(s')|$$

$$\leq \gamma \|v_S - v'_S\|,$$

using the non-expansiveness of the operator $P$ and Lemma 5. Hence we have that

$$\left\| \hat{M}Q_S - \min_{\boldsymbol{a}\in\mathcal{A}} \left[-\mathcal{R}_S(\cdot,\boldsymbol{a}) + \gamma\mathcal{P}^{\boldsymbol{a}}_{s's}v'_S\right] \right\| \leq \gamma \|v_S - v'_S\|. \tag{15}$$

Gathering the results of the three cases completes the proof of Theorem D.1.

## Proof of Theorem 3.1

The proposition is proven by proving three auxilary results:

**Lemma E.10.** *For any $v_S : \mathcal{S} \to \mathbb{R}$, the solution of the game $\mathcal{G}$ is a limit point of the convergent sequence $T^1 v_S, T^2 v_S, \ldots$.*

*Proof of Lemma E.10.* By Lemma E.9, we have that

$$\|T^{k+1} v_S - T^k v_S\| \leq \gamma \|T^k v_S - T^{k-1} v_S\| \leq \cdots \leq \gamma^k \|T v_S - v_S\|. \tag{16}$$

The result follows after considering the limit as $k \to \infty$ and using the boundedness of $\|T v_S - v_S\|$, we deduce that $T^k v_S, T^{k+1} v_S, \ldots,$ is a Cauchy sequence which concludes the proof. $\square$

**Proposition E.11.** *Denote by the finite game $\mathcal{G}^k$ in which Switcher, Adversary and $N$ MARL agents seek to maximise the following finite-horizon objectives:*

$$v_S^k(s|\boldsymbol{\pi}, \mathfrak{g}) = -\mathbb{E}_{\mathfrak{g}, \boldsymbol{\sigma}} \left[ \sum_{0 \leq t \leq k < \infty} \gamma^t \left( \mathcal{R}(s_t, \boldsymbol{a}_t) + c \cdot \mathbf{1}_{\mathcal{N}}(g(s_t)) \right) \Big| s_0 = s; \boldsymbol{a}_t \sim (\boldsymbol{\pi}, \boldsymbol{\sigma}) \right], \ \forall s \in \mathcal{S},$$

$$v_A^k(s|\boldsymbol{\pi}, \mathfrak{g}) = -\mathbb{E}_{\mathfrak{g}, \boldsymbol{\sigma}} \left[ \sum_{0 \leq t \leq k < \infty} \gamma^t \mathcal{R}(s_t, \boldsymbol{a}_t) \Big| s_0 = s; \boldsymbol{a}_t \sim (\boldsymbol{\pi}, \boldsymbol{\sigma}) \right], \ \forall s \in \mathcal{S},$$

$$v^k(s|\boldsymbol{\pi}, \mathfrak{g}) = \mathbb{E}_{\mathfrak{g}, \boldsymbol{\sigma}} \left[ \sum_{0 \leq t \leq k < \infty} \gamma^t \mathcal{R}(s_t, \boldsymbol{a}_t) \Big| s_0 = s; \boldsymbol{a}_t \sim (\boldsymbol{\pi}, \boldsymbol{\sigma}) \right], \ \forall s \in \mathcal{S},$$

*Let $\hat{\mathfrak{g}}$ and $\hat{\boldsymbol{\sigma}}, \hat{\boldsymbol{\pi}} \in \boldsymbol{\Pi}$ be the Markov policies generated by the process $T^1 v_S, T^2 v_S, \ldots$, then $\hat{\mathfrak{g}}$ and $\hat{\boldsymbol{\sigma}}, \hat{\boldsymbol{\pi}} \in \boldsymbol{\Pi}$ are optimal policies for $\mathcal{G}^k$.*

*Proof.* The proof is achieved using similar arguments as those presented in Theorem 2 of (Shapley, 1953) with some modifications. Denote by the *finite* game $\mathcal{G}^k$ of $k < \infty$ steps in which adversary (the $N$ agents) maximises (minimise) the following objective $v^k(s|\boldsymbol{\pi}, \mathfrak{g}) = -\mathbb{E}\left[\sum_{t=0}^k \gamma^t \{\mathcal{R}(s_t, a_t, b_t)\} \Big| s_0 = s\right]$ and Switcher maximises $v_C^k(s|\boldsymbol{\pi}, \mathfrak{g}) = -\mathbb{E}\left[\sum_{t=0}^k \gamma^t \left(\mathcal{R}(s_t, \boldsymbol{a}_t) + c \cdot \mathbf{1}_{\mathcal{N}}(g(s_t))\right) \Big| s_0 = s\right]$. Suppose in the game $\mathcal{G}^k$, Switcher is given a payoff of $-\mathcal{R}(s, \boldsymbol{a}) + \mathcal{P}_{s's}^{\boldsymbol{a}} v_S(s'|\boldsymbol{\pi}, \mathfrak{g})$ given Switcher policy $\mathfrak{g}$ and for any given $s \in \mathcal{S}$ and $\boldsymbol{a} \sim \boldsymbol{\pi}, \sigma$. Now the Markov policy $\mathfrak{g}$ guarantees Switcher a payoff of $v^k(s|\boldsymbol{\pi}, \mathfrak{g})$. Now in the game $\mathcal{G}^k$, after $n < \infty$ steps and using the policy $\mathfrak{g}$ gives Switcher an expected payoff of at least $v^k(s|\boldsymbol{\pi}, \mathfrak{g}) - \gamma^{n-1} \max_{\boldsymbol{a} \in \mathcal{A}} \mathcal{P}_{s's}^{\boldsymbol{a}} v^k(s'|\boldsymbol{\pi}, \mathfrak{g}) \leq v^k(s|\boldsymbol{\pi}, \mathfrak{g}) - \gamma^{n-1} \max_{s' \in \mathcal{S}} v^k(s'|\boldsymbol{\pi}, \mathfrak{g})$. Therefore, accounting for the $n$ steps, the total payoff for Switcher is at least $v^k(s|\boldsymbol{\pi}, \mathfrak{g}) - \gamma^{n-1} \max_{s' \in \mathcal{S}} v(s'|\boldsymbol{\pi}, \mathfrak{g}) - \sum_{t=0}^{n-1} \gamma^t v^{n-t}(s'|\boldsymbol{\pi}, \mathfrak{g}) = v^k(s|\boldsymbol{\pi}, \mathfrak{g}) - \gamma^{n-1} \max_{s' \in \mathcal{S}} v(s'|\boldsymbol{\pi}, \mathfrak{g}) - \gamma^n \frac{1-\gamma^n}{1-\gamma} \|v\| := \tilde{\boldsymbol{v}}^{k,n}(s|\boldsymbol{\pi}, \mathfrak{g})$. This expression holds for arbitrarily large values of $n$ in particular $\lim_{n \to \infty} \tilde{\boldsymbol{v}}^{k,n}(s|\boldsymbol{\pi}, \mathfrak{g}) = v^k(s|\boldsymbol{\pi}, \mathfrak{g})$ from which it follows that the policy $\mathfrak{g}$ is optimal for Switcher. Now by Assumption 3 it is easy to see for any fixed $K < \infty$, any policy which is optimal for Switcher in the game $\mathcal{G}$ is optimal for Switcher in a game $\tilde{\mathcal{G}}$ in which Switcher's objective is $v_S^k(s|\boldsymbol{\pi}, \mathfrak{g}) = -\mathbb{E}_{\mathfrak{g}, \boldsymbol{\sigma}} \left[\sum_{0 \leq t \leq k < \infty} \gamma^t \left(\mathcal{R}(s_t, \boldsymbol{a}_t) + c \cdot \mathbf{1}_{\mathcal{N}}(g(s_t))\right) \Big| s_0 = s; \boldsymbol{a}_t \sim (\boldsymbol{\pi}, \boldsymbol{\sigma})\right]$ with the game being otherwise identical. After using analogous arguments for Switcher and adversary, we deduce the result. $\square$

**Lemma E.12.** *The value of the game $\mathcal{G}$ is unique.*

*Proof.* Suppose there exists two values the game $\mathcal{G}$, $v_S'$ and $v_S$. Then, since each is a solution, we have by Lemma E.9 that each is a fixed point of the Bellman operator and hence $T v_S = v_S$ and $T v_S' = v_S'$. Hence, we have the following:

$$\|v_S - v_S'\| = \|T v_S - T v_S'| \leq \gamma \|v_S - v_S'\|, \tag{17}$$

whereafter, we immediately deduce that $v_S = v_S'$. $\square$

Summing up the above results we have succeeded in proving Theorem 3.1, moreover Proposition 3.2 follows as an immediate consequence. ∎

To prove the Theorem 3.3, we make use of the following result:

**Theorem E.13** (Theorem 1, pg 4 in (Jaakkola et al., 1994)). *Let $\Xi_t(s)$ be a random process that takes values in $\mathbb{R}^n$ and given by the following:*

$$\Xi_{t+1}(s) = (1 - \alpha_t(s)) \Xi_t(s) \alpha_t(s) L_t(s), \tag{18}$$

*then $\Xi_t(s)$ converges to 0 with probability 1 under the following conditions:*

i) $0 \leq \alpha_t \leq 1, \sum_t \alpha_t = \infty$ *and* $\sum_t \alpha_t < \infty$

ii) $\|\mathbb{E}[L_t|\mathcal{F}_t]\| \leq \gamma \|\Xi_t\|$, *with* $\gamma < 1$;

iii) $\mathrm{Var}\,[L_t|\mathcal{F}_t] \leq c(1 + \|\Xi_t\|^2)$ *for some* $c > 0$.

*Proof.* To prove the result, we show (i) - (iii) hold. Condition (i) holds by choice of learning rate. It therefore remains to prove (ii) - (iii). We first prove (ii). For this, we consider our variant of the Q-learning update rule:

$$
\begin{aligned}
\boldsymbol{Q}_{S,t+1}(s_t, \boldsymbol{a}_t, g|\cdot) = {} & \boldsymbol{Q}_{S,t}(s_t, \boldsymbol{a}_t, g|\cdot) \\
& + \alpha_t(s_t, \boldsymbol{a}_t) \Big[ \max \Big\{ \hat{\boldsymbol{\mathcal{M}}} Q_S(s_{\tau_k}, \boldsymbol{a}, g|\cdot), \mathcal{R}(s_{\tau_k}, \boldsymbol{a}, g) + \gamma \max_{\boldsymbol{a}' \in \mathcal{A}} Q_S(s_{t+1}, \boldsymbol{a}', g|\cdot) \Big\} \\
& \hspace{8cm} - \boldsymbol{Q}_t(s_t, \boldsymbol{a}_t, g|\cdot) \Big].
\end{aligned}
$$

After subtracting $\boldsymbol{Q}_S^*(s_t, \boldsymbol{a}_t, g|\cdot)$ from both sides and some manipulation we obtain that:

$$
\begin{aligned}
& \Xi_{t+1}(s_t, \boldsymbol{a}_t) \\
& = (1 - \alpha_t(s_t, \boldsymbol{a}_t)) \Xi_t(s_t, \boldsymbol{a}_t) \\
& \quad + \alpha_t(s_t, \boldsymbol{a}_t)) \left[ \max \Big\{ \hat{\boldsymbol{\mathcal{M}}} Q_S(s_{\tau_k}, \boldsymbol{a}, g|\cdot), \mathcal{R}_S(s_{\tau_k}, \boldsymbol{a}, g) + \gamma \max_{\boldsymbol{a}' \in \mathcal{A}} \boldsymbol{Q}_S(s', \boldsymbol{a}', g|\cdot) \Big\} - \boldsymbol{Q}_S^*(s_t, \boldsymbol{a}_t, g|\cdot) \right],
\end{aligned}
$$

where $\Xi_t(s_t, \boldsymbol{a}_t, g) := \boldsymbol{Q}_{S,t}(s_t, \boldsymbol{a}_t, g|\cdot) - Q_S^*(s_t, \boldsymbol{a}_t, g|\cdot)$.

Let us now define by

$$
L_t(s_{\tau_k}, \boldsymbol{a}, g) := \max \Big\{ \hat{\boldsymbol{\mathcal{M}}} Q_S(s_{\tau_k}, \boldsymbol{a}, g|\cdot), \mathcal{R}_S(s_{\tau_k}, \boldsymbol{a}, g) + \gamma \max_{\boldsymbol{a}' \in \mathcal{A}} \boldsymbol{Q}_S(s', \boldsymbol{a}', g|\cdot) \Big\} - \boldsymbol{Q}_S^*(s_t, \boldsymbol{a}, g|\cdot).
$$

Then

$$\Xi_{t+1}(s_t, \boldsymbol{a}_t, g) = (1 - \alpha_t(s_t, \boldsymbol{a}_t)) \Xi_t(s_t, \boldsymbol{a}_t, g) + \alpha_t(s_t, \boldsymbol{a}_t)) \left[ L_t(s_{\tau_k}, \boldsymbol{a}, g) \right]. \tag{19}$$

We now observe that

$$
\begin{aligned}
& \mathbb{E}\left[L_t(s_{\tau_k}, \boldsymbol{a}, g)|\mathcal{F}_t\right] \\
& = \sum_{s' \in \mathcal{S}} P(s'; a, s_{\tau_k}) \max \Big\{ \hat{\boldsymbol{\mathcal{M}}} Q_S(s_{\tau_k}, \boldsymbol{a}, g|\cdot), \mathcal{R}_S(s_{\tau_k}, \boldsymbol{a}, g) + \gamma \max_{\boldsymbol{a}' \in \mathcal{A}} \boldsymbol{Q}_S(s', \boldsymbol{a}', g|\cdot) \Big\} \\
& \hspace{10cm} - \boldsymbol{Q}_S^*(s_{\tau_k}, a, g|\cdot) \\
& = T \boldsymbol{Q}_{S,t}(s, \boldsymbol{a}, g|\cdot) - \boldsymbol{Q}_S^*(s, \boldsymbol{a}, g). \tag{20}
\end{aligned}
$$

Now, using the fixed point property that implies $\boldsymbol{Q}^* = T\boldsymbol{Q}^*$, we find that

$$
\begin{aligned}
\mathbb{E}\left[L_t(s_{\tau_k}, \boldsymbol{a}, g)|\mathcal{F}_t\right] & = T\boldsymbol{Q}_{S,t}(s, \boldsymbol{a}, g|\cdot) - T\boldsymbol{Q}_S^*(s, \boldsymbol{a}, g|\cdot) \\
& \leq \|T\boldsymbol{Q}_{S,t} - T\boldsymbol{Q}^*\| \\
& \leq \gamma \|\boldsymbol{Q}_{S,t} - \boldsymbol{Q}^*\|_\infty = \gamma \|\Xi_t\|_\infty. \tag{21}
\end{aligned}
$$

using the contraction property of $T$ established in Lemma E.9. This proves (ii).

We now prove iii), that is

$$\text{Var}\left[L_t|\mathcal{F}_t\right] \leq c(1 + \|\Xi_t\|^2). \tag{22}$$

Now by equation 20 we have that

$$\begin{aligned}
\text{Var}\left[L_t|\mathcal{F}_t\right] &= \text{Var}\left[\max\left\{\hat{\mathcal{M}}Q_S(s_{\tau_k}, \boldsymbol{a}, g|\cdot), \mathcal{R}_S(s_{\tau_k}, \boldsymbol{a}, g) + \gamma\max_{\boldsymbol{a}'\in\mathcal{A}} \boldsymbol{Q}_S(s', \boldsymbol{a}', g|\cdot)\right\} - \boldsymbol{Q}_S^*(s_t, \boldsymbol{a}, g|\cdot)\right] \\
&= \mathbb{E}\left[\left(\max\left\{\hat{\mathcal{M}}Q_S(s_{\tau_k}, \boldsymbol{a}, g|\cdot), \mathcal{R}_S(s_{\tau_k}, \boldsymbol{a}, g) + \gamma\max_{\boldsymbol{a}'\in\mathcal{A}} \boldsymbol{Q}_S(s', \boldsymbol{a}', g|\cdot)\right\}\right.\right. \\
&\qquad\qquad \left.\left. - \boldsymbol{Q}_S^*(s_t, \boldsymbol{a}, g|\cdot) - (T\boldsymbol{Q}_{S,t}(s, \boldsymbol{a}, g|\cdot) - \boldsymbol{Q}_S^*(s, \boldsymbol{a}, g|\cdot))\right)^2\right] \\
&= \mathbb{E}\left[\left(\max\left\{\hat{\mathcal{M}}Q_S(s_{\tau_k}, \boldsymbol{a}, g|\cdot), \mathcal{R}_S(s_{\tau_k}, \boldsymbol{a}, g) + \gamma\max_{\boldsymbol{a}'\in\mathcal{A}} Q_S(s', \boldsymbol{a}', g|\cdot)\right\} - T\boldsymbol{Q}_{S,t}(s, \boldsymbol{a}, g|\cdot)\right)^2\right] \\
&= \text{Var}\left[\max\left\{\hat{\mathcal{M}}Q_S(s_{\tau_k}, \boldsymbol{a}, g|\cdot), \mathcal{R}_S(s_{\tau_k}, \boldsymbol{a}, g) + \gamma\max_{\boldsymbol{a}'\in\mathcal{A}} Q_S(s', \boldsymbol{a}', g|\cdot)\right\} - T\boldsymbol{Q}_{S,t}(s, \boldsymbol{a}, g|\cdot))\right] \\
&\leq c(1 + \|\Xi_t\|^2),
\end{aligned}$$

for some $c > 0$ where the last line follows due to the boundedness of $Q$ (which follows from Assumptions 2 and 4). This concludes the proof of the Theorem. $\qquad\square$

Theorem 3.3 proves the convergence of MARTA to the solution to the game $\mathcal{G}$ (and that such a solution exists). In particular, the following result follows a consequence of Theorem 3.3 and the uniqueness of the value function solution of $\mathcal{G}$:

## F. Convergence of MARTA with Linear Function Approximators

In RL, an important consideration is the use of function approximators for the functions being learned during the learning process. We now extend the convergence results established in the previous section to linear function approximators. Linear function approximators are an important class due to their simplicity and convexity properties that do not suffer from issues such as convergence to suboptimal stationary points. Moreover, the following analysis is an important stepping stone for proving analogous results with other function approximator classes.

In addition to Assumptions 1 - 3, the results of this section are built under the following assumptions: Assumption 4. For any positive scalar $c$, there exists a scalar $\kappa_c$ such that for all $s \in \mathcal{S}$ and for any $t \in \mathbb{N}$ we have: $\mathbb{E}\left[1 + \|s_t\|^c|s_0 = s\right] \leq \kappa_c(1 + \|s\|^c)$.

Assumption 5. There exists scalars $C_1$ and $c_1$ such that for any function $v$ satisfying $|v(s)| \leq C_2(1 + \|s\|^{c_2})$ for some scalars $c_2$ and $C_2$ we have that: $\sum_{t=0}^{\infty} |\mathbb{E}\left[v(s_t)|s_0 = s\right] - \mathbb{E}[v(s_0)]| \leq C_1 C_2(1 + \|s_0\|^{c_1 c_2})$.

Assumption 6. There exists scalars $c$ and $C$ such that for any $s \in \mathcal{S}$ we have that $|R(s, \cdot)| \leq C(1 + \|s\|^c)$.

Theorem 3.4 is proven using a set of results that we now establish. First we prove the following bound holds:

**Lemma F.1.** *For any $\boldsymbol{Q} \in L_2$ we have that*

$$\|\mathfrak{F}\boldsymbol{Q} - \boldsymbol{Q}'\| \leq \gamma \|\boldsymbol{Q} - \boldsymbol{Q}'\|, \tag{23}$$

*so that the operator $\mathfrak{F}$ is a contraction.*

*Proof.* Now, we first note that by result iv) in the proof of Lemma E.9, we deduced that for any $\boldsymbol{Q}, v \in L_2$ we have that

$$\|\mathcal{M}\boldsymbol{Q} - [\mathcal{R}(\cdot, \boldsymbol{a}) + \gamma\mathcal{P}^{\boldsymbol{a}}v']\| \leq \gamma \|v - v'\|.$$

where $\mathcal{P}^{\boldsymbol{a}}_{ss'} =: \sum_{s'\in\mathcal{S}} P(s'; \boldsymbol{a}, s)$.

Hence, using the contraction property of $\mathcal{M}$ and results i)-iv) of Lemma E.9, we readily deduce the following bound:

$$\max\left\{\left\|\mathcal{M}\boldsymbol{Q} - \hat{\boldsymbol{Q}}\right\|, \left\|\mathcal{M}\boldsymbol{Q} - \mathcal{M}\hat{\boldsymbol{Q}}\right\|\right\} \leq \gamma\left\|\boldsymbol{Q} - \hat{\boldsymbol{Q}}\right\|. \tag{24}$$

We now observe that $\mathfrak{F}$ is a contraction. Indeed, since for any $\boldsymbol{Q}, \boldsymbol{Q}' \in L_2$ we have that:

$$
\begin{aligned}
&\|\mathfrak{F}\boldsymbol{Q} - \mathfrak{F}\boldsymbol{Q}'\| \\
&= \left\|\mathcal{R} + \gamma P \min\{\hat{\mathcal{M}}Q, Q\} - \left(\mathcal{R} + \gamma P \min\{\hat{\mathcal{M}}Q', Q'\}\right)\right\| \\
&= \gamma\left\|P\min\{\hat{\mathcal{M}}Q, Q\} - P\min\{\hat{\mathcal{M}}Q', Q'\}\right\| \\
&\leq \gamma\max\left\{\|\boldsymbol{Q} - \boldsymbol{Q}'\|, \gamma\|\boldsymbol{Q} - \boldsymbol{Q}'\|\right\} \\
&= \gamma\|\boldsymbol{Q} - \boldsymbol{Q}'\|
\end{aligned}
$$

using the Cauchy-Schwarz inequality, equation 24 and again using the non-expansiveness of $P$. $\qquad\square$

We next show that the following two bounds hold:

**Lemma F.2.** *For any $\boldsymbol{Q} \in \mathcal{V}$ we have that*

i) $$\left\|\Pi\mathfrak{F}\boldsymbol{Q} - \Pi\mathfrak{F}\bar{\boldsymbol{Q}}\right\| \leq \gamma\left\|\boldsymbol{Q} - \bar{\boldsymbol{Q}}\right\|,$$

ii) $$\left\|\Phi r^\star - \boldsymbol{Q}^\star\right\| \leq \frac{1}{\sqrt{1-\gamma^2}}\left\|\Pi\boldsymbol{Q}^\star - \boldsymbol{Q}^\star\right\|.$$

*Proof.* The first result is straightforward since as $\Pi$ is a projection it is non-expansive and hence:

$$\left\|\Pi\mathfrak{F}\boldsymbol{Q} - \Pi\mathfrak{F}\bar{\boldsymbol{Q}}\right\| \leq \left\|\mathfrak{F}\boldsymbol{Q} - \mathfrak{F}\bar{\boldsymbol{Q}}\right\| \leq \gamma\left\|\boldsymbol{Q} - \bar{\boldsymbol{Q}}\right\|,$$

using the contraction property of $\mathfrak{F}$. This proves i). For ii), we note that by the orthogonality property of projections we have that $\langle\Phi r^\star - \Pi\boldsymbol{Q}^\star, \Phi r^\star - \Pi\boldsymbol{Q}^\star\rangle = 0$, hence we observe that:

$$
\begin{aligned}
\|\Phi r^\star - \boldsymbol{Q}^\star\|^2 &= \|\Phi r^\star - \Pi\boldsymbol{Q}^\star\|^2 + \|\boldsymbol{Q}^* - \Pi\boldsymbol{Q}^\star\|^2 \\
&= \|\Pi\mathfrak{F}\Phi r^\star - \Pi\boldsymbol{Q}^\star\|^2 + \|\boldsymbol{Q}^* - \Pi\boldsymbol{Q}^\star\|^2 \\
&\leq \|\mathfrak{F}\Phi r^\star - \boldsymbol{Q}^\star\|^2 + \|\boldsymbol{Q}^* - \Pi\boldsymbol{Q}^\star\|^2 \\
&= \|\mathfrak{F}\Phi r^\star - \mathfrak{F}\boldsymbol{Q}^\star\|^2 + \|\boldsymbol{Q}^* - \Pi\boldsymbol{Q}^\star\|^2 \\
&\leq \gamma^2\|\Phi r^\star - \boldsymbol{Q}^\star\|^2 + \|\boldsymbol{Q}^* - \Pi\boldsymbol{Q}^\star\|^2,
\end{aligned}
$$

after which we readily deduce the desired result. $\qquad\square$

**Lemma F.3.** *Define the operator $H$ by the following:*

$$H\boldsymbol{Q}(s, a, b) = \begin{cases} \hat{\mathcal{M}}\boldsymbol{Q}(s, \boldsymbol{a}), & \text{if } \hat{\mathcal{M}}\boldsymbol{Q}(s, \boldsymbol{a}) < \Phi r^\star \\ \boldsymbol{Q}(s, \boldsymbol{a}), & \text{otherwise}, \end{cases}$$

*where we define $\tilde{\mathfrak{F}}$ by: $\tilde{\mathfrak{F}}\boldsymbol{Q} := \mathcal{R} + \gamma PH\boldsymbol{Q}$.*

*For any $\boldsymbol{Q}, \bar{\boldsymbol{Q}} \in L_2$ we have that*

$$\left\|\tilde{\mathfrak{F}}\boldsymbol{Q} - \tilde{\mathfrak{F}}\bar{\boldsymbol{Q}}\right\| \leq \gamma\left\|\boldsymbol{Q} - \bar{\boldsymbol{Q}}\right\| \tag{25}$$

*and hence $\tilde{\mathfrak{F}}$ is a contraction mapping.*

*Proof.* Using equation 24, we now observe that

$$\left\| \tilde{\mathfrak{F}} \boldsymbol{Q} - \tilde{\mathfrak{F}} \bar{\boldsymbol{Q}} \right\| = \left\| \mathcal{R} + \gamma P H \boldsymbol{Q} - (\mathcal{R} + \gamma P H \bar{\boldsymbol{Q}}) \right\|$$
$$\leq \gamma \left\| H \boldsymbol{Q} - H \bar{\boldsymbol{Q}} \right\|$$
$$\leq \gamma \left\| \max \left\{ \hat{\mathcal{M}} \boldsymbol{Q} - \hat{\mathcal{M}} \bar{\boldsymbol{Q}}, \hat{\mathcal{M}} \boldsymbol{Q} - \bar{\boldsymbol{Q}}, \hat{\mathcal{M}} \bar{\boldsymbol{Q}} - \boldsymbol{Q} \right\} \right\|$$
$$\leq \gamma \max \left\{ \left\| \hat{\mathcal{M}} \boldsymbol{Q} - \hat{\mathcal{M}} \bar{\boldsymbol{Q}} \right\|, \left\| \hat{\mathcal{M}} \boldsymbol{Q} - \bar{\boldsymbol{Q}} \right\|, \left\| \hat{\mathcal{M}} \bar{\boldsymbol{Q}} - \boldsymbol{Q} \right\| \right\}$$
$$\leq \gamma \max \left\{ \gamma \left\| \boldsymbol{Q} - \bar{\boldsymbol{Q}} \right\|, \left\| \boldsymbol{Q} - \bar{\boldsymbol{Q}} \right\| \right\}$$
$$= \gamma \left\| \boldsymbol{Q} - \bar{\boldsymbol{Q}} \right\|,$$

again using equation 24 and the non-expansive property of $P$. $\qquad\square$

**Lemma F.4.** *Define by* $\tilde{\boldsymbol{Q}} := \mathcal{R} + \gamma P v^{\tilde{\sigma}}$ *where*

$$v^{\tilde{\sigma}}(s) := \min \left\{ \hat{\mathcal{M}} \boldsymbol{Q}^{\tilde{\sigma}}(s, \boldsymbol{a}), \mathcal{R}(s, \boldsymbol{a}) + \gamma \mathbb{E}_{s' \sim P} \left[ v^{\tilde{\sigma}}(s') \right] \right\}, \tag{26}$$

*then* $\tilde{\boldsymbol{Q}}$ *is a fixed point of* $\tilde{\mathfrak{F}} \tilde{\boldsymbol{Q}}$, *that is* $\tilde{\mathfrak{F}} \tilde{\boldsymbol{Q}} = \tilde{\boldsymbol{Q}}$.

*Proof.* We begin by observing that

$$H \tilde{\boldsymbol{Q}}(s, \boldsymbol{a}) = H \left( \mathcal{R}(s, \boldsymbol{a}) + \gamma \mathcal{P}_{ss'}^{\boldsymbol{a}} v^{\tilde{\sigma}}(s') \right)$$
$$= \begin{cases} \hat{\mathcal{M}} \boldsymbol{Q}(s, \boldsymbol{a}), & \text{if } \hat{\mathcal{M}} \boldsymbol{Q}(s, \boldsymbol{a}) > \Phi r^{\star} \\ \boldsymbol{Q}(s, \boldsymbol{a}), & \text{otherwise,} \end{cases}$$
$$= \begin{cases} \hat{\mathcal{M}} \boldsymbol{Q}(s, \boldsymbol{a}), & \text{if } \hat{\mathcal{M}} \boldsymbol{Q}(s, \boldsymbol{a}) > \Phi r^{\star}, \\ \mathcal{R}(s, \boldsymbol{a}) + \gamma P v^{\tilde{\sigma}}, & \text{otherwise,} \end{cases}$$
$$= v^{\tilde{\sigma}}(s).$$

Hence,

$$\tilde{\mathfrak{F}} \tilde{\boldsymbol{Q}} = \mathcal{R} + \gamma P H \tilde{\boldsymbol{Q}} = \mathcal{R} + \gamma P v^{\tilde{\sigma}} = \tilde{\boldsymbol{Q}}. \tag{27}$$

which proves the result. $\qquad\square$

**Lemma F.5.** *The following bound holds:*

$$\mathbb{E} \left[ v^{\hat{\sigma}}(s_0) \right] - \mathbb{E} \left[ v^{\tilde{\sigma}}(s_0) \right] \leq 2 \left[ (1 - \gamma) \sqrt{(1 - \gamma^2)} \right]^{-1} \left\| \Pi \boldsymbol{Q}^{\star} - \boldsymbol{Q}^{\star} \right\|. \tag{28}$$

*Proof.* By definitions of $v^{\hat{\sigma}}$ and $v^{\tilde{\sigma}}$ (c.f equation 26) and using Jensen's inequality and the stationarity property we have that,

$$\mathbb{E} \left[ v^{\hat{\sigma}}(s_0) \right] - \mathbb{E} \left[ v^{\tilde{\sigma}}(s_0) \right] = \mathbb{E} \left[ P v^{\hat{\sigma}}(s_0) \right] - \mathbb{E} \left[ P v^{\tilde{\sigma}}(s_0) \right]$$
$$\leq \left| \mathbb{E} \left[ P v^{\hat{\sigma}}(s_0) \right] - \mathbb{E} \left[ P v^{\tilde{\sigma}}(s_0) \right] \right|$$
$$\leq \left\| P v^{\hat{\sigma}} - P v^{\tilde{\sigma}} \right\|. \tag{29}$$

Now recall that $\tilde{\boldsymbol{Q}} := \mathcal{R} + \gamma P v^{\tilde{\sigma}}$ and $\boldsymbol{Q}^{\star} := \mathcal{R} + \gamma P v^{\sigma\star}$, using these expressions in equation 29 we find that

$$\mathbb{E} \left[ v^{\hat{\sigma}}(s_0) \right] - \mathbb{E} \left[ v^{\tilde{\sigma}}(s_0) \right] \leq \frac{1}{\gamma} \left\| \tilde{\boldsymbol{Q}} - \boldsymbol{Q}^{\star} \right\|.$$

Moreover, by the triangle inequality and using the fact that $\mathfrak{F}(\Phi r^\star) = \tilde{\mathfrak{F}}(\Phi r^\star)$ and that $\mathfrak{F}Q^\star = Q^\star$ and $\mathfrak{F}\tilde{Q} = \tilde{Q}$ (c.f. equation 28) we have that

$$\left\| \tilde{Q} - Q^\star \right\| \leq \left\| \tilde{Q} - \mathfrak{F}(\Phi r^\star) \right\| + \left\| Q^\star - \tilde{\mathfrak{F}}(\Phi r^\star) \right\|$$

$$\leq \gamma \left\| \tilde{Q} - \Phi r^\star \right\| + \gamma \left\| Q^\star - \Phi r^\star \right\|$$

$$\leq 2\gamma \left\| \tilde{Q} - \Phi r^\star \right\| + \gamma \left\| Q^\star - \tilde{Q} \right\|,$$

which gives the following bound:

$$\left\| \tilde{Q} - Q^\star \right\| \leq 2\gamma \left(1 - \gamma\right)^{-1} \left\| \tilde{Q} - \Phi r^\star \right\|,$$

from which, using Lemma F.2, we deduce that $\left\| \tilde{Q} - Q^\star \right\| \leq 2\gamma \left[ (1 - \gamma)\sqrt{(1 - \gamma^2)} \right]^{-1} \|\Pi Q^\star - Q^\star\|$, after which by equation 30, we finally obtain

$$\mathbb{E}\left[ v^{\hat{\sigma}}(s_0) \right] - \mathbb{E}\left[ v^{\tilde{\sigma}}(s_0) \right] \leq 2 \left[ (1 - \gamma)\sqrt{(1 - \gamma^2)} \right]^{-1} \|\Pi Q^\star - Q^\star\|,$$

as required. $\qquad\square$

Let us rewrite the update in the following way:

$$r_{t+1} = r_t + \gamma_t \Xi(w_t, r_t),$$

where the function $\Xi : \mathbb{R}^{2d} \times \mathbb{R}^p \to \mathbb{R}^p$ is given by:

$$\Xi(w, r) := \phi(s) \left( \mathcal{R}(s, \cdot) + \gamma \min\{(\Phi r)(s'), \hat{\mathcal{M}}(\Phi r)(s')\} - (\Phi r)(s) \right),$$

for any $w \equiv (s, s') \in \mathcal{S}^2$ and for any $r \in \mathbb{R}^p$. Let us also define the function $\boldsymbol{\Xi} : \mathbb{R}^p \to \mathbb{R}^p$ by the following:

$$\boldsymbol{\Xi}(r) := \mathbb{E}_{w_0 \sim (\mathbb{P}, \mathbb{P})} \left[ \Xi(w_0, r) \right]; w_0 := (s_0, z_1).$$

**Lemma F.6.** *The following statements hold for all $z \in \{0, 1\} \times \mathcal{S}$:*

   *i) $(r - r^\star)\boldsymbol{\Xi}_k(r) < 0, \qquad \forall r \neq r^\star$,*

  *ii) $\boldsymbol{\Xi}_k(r^\star) = 0$.*

*Proof.* To prove the statement, we first note that each component of $\boldsymbol{\Xi}_k(r)$ admits a representation as an inner product, indeed:

$$\boldsymbol{\Xi}_k(r) = \mathbb{E}\left[ \phi_k(s_0)(\mathcal{R}(s_0, \boldsymbol{a}_0) + \gamma \min\{(\Phi r)(s_1), \hat{\mathcal{M}}(\Phi r)(s_1)\} - (\Phi r)(s_0) \right]$$

$$= \mathbb{E}\left[ \phi_k(s_0)(\mathcal{R}(s_0, \boldsymbol{a}_0) + \gamma \mathbb{E}\left[ \min\{(\Phi r)(s_1), \hat{\mathcal{M}}(\Phi r)(s_1)\} | x_0 \right] - (\Phi r)(s_0) \right]$$

$$= \mathbb{E}\left[ \phi_k(s_0)(\mathcal{R}(s_0, \boldsymbol{a}_0) + \gamma P \min\{(\Phi r), \hat{\mathcal{M}}(\Phi r)\}(s_0) - (\Phi r)(s_0) \right]$$

$$= \langle \phi_k, \mathfrak{F}\Phi r - \Phi r \rangle,$$

using the iterated law of expectations and the definitions of $P$ and $\mathfrak{F}$.

We now are in a position to prove (i). Indeed, we now observe the following:

$$(r - r^\star)\boldsymbol{\Xi}_k(r) = \sum_{l=1} (r(l) - r^\star(l)) \langle \phi_l, \mathfrak{F}\Phi r - \Phi r \rangle$$

$$= \langle \Phi r - \Phi r^\star, \mathfrak{F}\Phi r - \Phi r \rangle$$

$$= \langle \Phi r - \Phi r^\star, (\mathbf{1} - \Pi)\mathfrak{F}\Phi r + \Pi\mathfrak{F}\Phi r - \Phi r \rangle$$

$$= \langle \Phi r - \Phi r^\star, \Pi\mathfrak{F}\Phi r - \Phi r \rangle,$$

where in the last step we used the orthogonality of $(\mathbf{1} - \Pi)$. We now recall that $\Pi\mathfrak{F}\Phi r^\star = \Phi r^\star$ since $\Phi r^\star$ is a fixed point of $\Pi\mathfrak{F}$. Additionally, using Lemma F.2 we observe that $\|\Pi\mathfrak{F}\Phi r - \Phi r^\star\| \leq \gamma\|\Phi r - \Phi r^\star\|$. With this we now find that

$$
\begin{aligned}
&\langle \Phi r - \Phi r^\star, \Pi\mathfrak{F}\Phi r - \Phi r\rangle \\
&= \langle \Phi r - \Phi r^\star, (\Pi\mathfrak{F}\Phi r - \Phi r^\star) + \Phi r^\star - \Phi r\rangle \\
&\leq \|\Phi r - \Phi r^\star\| \, \|\Pi\mathfrak{F}\Phi r - \Phi r^\star\| - \|\Phi r^\star - \Phi r\|^2 \\
&\leq (\gamma - 1)\, \|\Phi r^\star - \Phi r\|^2,
\end{aligned}
$$

which is negative since $\gamma < 1$ which completes the proof of part i).

The proof of part ii) is straightforward since we readily observe that

$$
\Xi_k(r^\star) = \langle \phi_l, \mathfrak{F}\Phi r^\star - \Phi r\rangle = \langle \phi_l, \Pi\mathfrak{F}\Phi r^\star - \Phi r\rangle = 0,
$$

as required and from which we deduce the result. $\qquad\square$

To prove the theorem, we make use of a special case of the following result:

**Theorem F.7** (Th. 17, p. 239 in (Benveniste et al., 2012))**.** *Consider a stochastic process $r_t : \mathbb{R} \times \{\infty\} \times \Omega \to \mathbb{R}^k$ which takes an initial value $r_0$ and evolves according to the following:*

$$
r_{t+1} = r_t + \alpha\Xi(s_t, r_t), \tag{30}
$$

*for some function $s : \mathbb{R}^{2d} \times \mathbb{R}^k \to \mathbb{R}^k$ and where the following statements hold:*

1. *$\{s_t | t = 0, 1, \ldots\}$ is a stationary, ergodic Markov process taking values in $\mathbb{R}^{2d}$*

2. *For any positive scalar $q$, there exists a scalar $\mu_q$ such that $\mathbb{E}\left[1 + \|s_t\|^q | s \equiv s_0\right] \leq \mu_q\left(1 + \|s\|^q\right)$*

3. *The step size sequence satisfies the Robbins-Monro conditions, that is $\sum_{t=0}^\infty \alpha_t = \infty$ and $\sum_{t=0}^\infty \alpha_t^2 < \infty$*

4. *There exists scalars $d$ and $q$ such that $\|\Xi(w, r)\| \leq d\left(1 + \|w\|^q\right)\left(1 + \|r\|\right)$*

5. *There exists scalars $d$ and $q$ such that $\sum_{t=0}^\infty \|\mathbb{E}\left[\Xi(w_t, r) | z_0 \equiv z\right] - \mathbb{E}\left[\Xi(w_0, r)\right]\| \leq d\left(1 + \|w\|^q\right)\left(1 + \|r\|\right)$*

6. *There exists a scalar $d > 0$ such that $\|\mathbb{E}[\Xi(w_0, r)] - \mathbb{E}[\Xi(w_0, \bar{r})]\| \leq d\|r - \bar{r}\|$*

7. *There exists scalars $d > 0$ and $q > 0$ such that $\sum_{t=0}^\infty \|\mathbb{E}\left[\Xi(w_t, r) | w_0 \equiv w\right] - \mathbb{E}\left[\Xi(w_0, \bar{r})\right]\| \leq c\|r - \bar{r}\|\left(1 + \|w\|^q\right)$*

8. *There exists some $r^\star \in \mathbb{R}^k$ such that $\Xi(r)(r - r^\star) < 0$ for all $r \neq r^\star$ and $\bar{s}(r^\star) = 0$.*

*Then $r_t$ converges to $r^\star$ almost surely.*

In order to apply the Theorem F.7, we show that conditions 1 - 7 are satisfied.

*Proof.* Conditions 1-2 are true by assumption while condition 3 can be made true by choice of the learning rates. Therefore it remains to verify conditions 4-7 are met.

To prove 4, we observe that

$$
\begin{aligned}
\|\Xi(w, r)\| &= \left\| \phi(s)\left(\mathcal{R}(s, \cdot) + \gamma\min\{(\Phi r)(s'), \hat{\mathcal{M}}(\Phi r)(s')\} - (\Phi r)(s)\right)\right\| \\
&\leq \|\phi(s)\| \left\| \mathcal{R}(s, \cdot) + \gamma\left(\|\phi(s')\| \, \|r\| + \hat{\mathcal{M}}\Phi(s')\right)\right\| + \|\phi(s)\| \, \|r\| \\
&\leq \|\phi(s)\| \left(\|\mathcal{R}(s, \cdot)\| + \gamma\|\hat{\mathcal{M}}\Phi(s')\|\right) + \|\phi(s)\|\left(\gamma\|\phi(s')\| + \|\phi(s)\|\right)\|r\|.
\end{aligned}
$$

Now using the definition of $\mathcal{M}$, we readily observe that $\|\mathcal{M}\Phi(s')\| \leq \|c\|_\infty + \|\mathcal{R}\| + \gamma\|\mathcal{P}^\sigma_{s's_t}\Phi\| \leq \|c\|_\infty + \|\mathcal{R}\| + \gamma\|\Phi\|$ using the non-expansiveness of $P$.

Hence, we lastly deduce that

$$\|\Xi(w,r)\| \le \|\phi(s)\| \left(\|\mathcal{R}(s,\cdot)\| + \gamma\|\hat{\boldsymbol{\mathcal{M}}}\Phi(s')\|\right) + \|\phi(s)\| \left(\gamma\|\phi(s')\| + \|\phi(s)\|\right)\|r\|$$

$$\le \|\phi(s)\| \left(\|\mathcal{R}(s,\cdot)\| + \gamma\left(\|c\|_\infty + \|\mathcal{R}\| + |\phi|\right)\right) + \|\phi(s)\| \left(\gamma\|\phi(s')\| + \|\phi(s)\|\right)\|r\|,$$

we then easily deduce the result using the boundedness of $\phi$ and $\mathcal{R}$.

Now we observe the following Lipschitz condition on $\Xi$:

$$\|\Xi(w,r) - \Xi(w,\bar{r})\|$$

$$= \left\|\phi(s)\left(\gamma\min\{(\Phi r)(s'), \hat{\boldsymbol{\mathcal{M}}}\Phi(s')\} - \gamma\min\{(\Phi\bar{r})(s'), \hat{\boldsymbol{\mathcal{M}}}\Phi(s')\}\right) - \left((\Phi r)(s) - \Phi\bar{r}(s)\right)\right\|$$

$$\le \gamma\|\phi(s)\| \left\|\min\left\{\phi'(s')r, \hat{\boldsymbol{\mathcal{M}}}\Phi'(s')\right\} - \min\left\{(\phi'(s')\bar{r}), \hat{\boldsymbol{\mathcal{M}}}\Phi'(s')\right\}\right\| + \|\phi(s)\| \|\phi'(s)r - \phi(s)\bar{r}\|$$

$$\le \gamma\|\phi(s)\| \|\phi'(s')r - \phi'(s')\bar{r}\| + \|\phi(s)\| \|\phi'(s)r - \phi'(s)\bar{r}\|$$

$$\le \|\phi(s)\| \left(\gamma\|\phi'(s')\| + \|\phi'(s)\|\right)\|r - \bar{r}\|$$

$$\le \text{const.} \|r - \bar{r}\|,$$

using Cauchy-Schwarz inequality and that for any scalars $a, b, c$ we have that $|\max\{a,b\} - \max\{b,c\}| \le |a - c|$ and $|\min\{a,b\} - \min\{b,c\}| \le |a - c|$, which proves Part 6.

Using Assumptions 3 and 4, we therefore deduce that

$$\sum_{t=0}^{\infty} \|\mathbb{E}[\Xi(w,r) - \Xi(w,\bar{r})|w_0 = w] - \mathbb{E}[\Xi(w_0,r) - \Xi(w_0,\bar{r})]\| \le \text{const.} \|r - \bar{r}\| (1 + \|w\|^l). \tag{31}$$

which proves Part 7.

Part 2 is assured by Lemma F.2 while Part 4 (and hence Part 5) is assured by Lemma F.5 and lastly Part 8 is assured by Lemma F.6. This result completes the proof of Theorem 3.4. $\qquad\square$

## Proof of Proposition E.8

*Proof.* We begin by re-expressing the *activation times* at which Switcher agent activates the Adversary. In particular, an activation time $\tau_k$ is defined recursively $\tau_k = \inf\{t > \tau_{k-1}|s_t \in A, \tau_k \in \mathcal{F}_t\}$ where $A = \{s \in \mathcal{S}, g(s_t) = 1\}$. The proof is given by deriving a contradiction. Therefore suppose that $\hat{\boldsymbol{\mathcal{M}}}v_S(s_{\tau_k}) > v_S(s_{\tau_k})$ and suppose that the activation time $\tau_1' > \tau_1$ is an optimal activation time. Construct Switcher $g'$ and $\tilde{g}$ policy activation times by $(\tau_0', \tau_1', \ldots,)$ and $g'^2$ policy by $(\tau_0', \tau_1, \ldots)$ respectively. Define by $l = \inf\{t > 0; \hat{\boldsymbol{\mathcal{M}}}\psi(s_t = \psi(s_t)\}$ and $m = \sup\{t; t < \tau_1'\}$. By construction we have that

$$v_S^{\boldsymbol{\pi},g'}(s)$$

$$= \mathbb{E}\left[\mathcal{R}_S(s_0, \boldsymbol{a}_0) + \mathbb{E}\left[\ldots + \gamma^{l-1}\mathbb{E}\left[\mathcal{R}_S(s_{\tau_1-1}, \boldsymbol{a}_{\tau_1-1}) + \ldots + \gamma^{m-l-1}\mathbb{E}\left[\mathcal{R}_S(s_{\tau_1'-1}, \boldsymbol{a}_{\tau_1'-1}) + \gamma\hat{\boldsymbol{\mathcal{M}}}v_S^{\boldsymbol{\pi},g'}(s')\right]\right]\right]\right]$$

$$< \mathbb{E}\left[\mathcal{R}_S(s_0, \boldsymbol{a}_0) + \mathbb{E}\left[\ldots + \gamma^{l-1}\mathbb{E}\left[\mathcal{R}_S(s_{\tau_1-1}, \boldsymbol{a}_{\tau_1-1}) + \gamma\hat{\boldsymbol{\mathcal{M}}}v_S^{\boldsymbol{\pi},g'}(s_{\tau_1})\right]\right]\right]$$

We now use the following observation $\mathbb{E}\left[\mathcal{R}_S(s_{\tau_1-1}, \boldsymbol{a}_{\tau_1-1}) + \gamma\hat{\boldsymbol{\mathcal{M}}}v_S^{\boldsymbol{\pi},g'}(s_{\tau_1})\right]$

$$\ge \min\left\{\hat{\boldsymbol{\mathcal{M}}}v_S^{\boldsymbol{\pi},g'}(s_{\tau_1}), \max_{\boldsymbol{a}_{\tau_1}\in\mathcal{A}}\left[\mathcal{R}_S(s_{\tau_1}, \boldsymbol{a}_{\tau_1}) + \gamma\sum_{s'\in\mathcal{S}}P(s'; \boldsymbol{a}_{\tau_1}, s_{\tau_1})v_S^{\boldsymbol{\pi},g}(s')\right]\right\}.$$

Using this we deduce that

$$v_S^{\boldsymbol{\pi},g'}(s > \mathbb{E}\left[\mathcal{R}_S(s_0, \boldsymbol{a}_0) + \mathbb{E}\left[\ldots\right.\right.$$

$$\left.\left. + \gamma^{l-1}\mathbb{E}\left[\mathcal{R}_S(s_{\tau_1-1}, \boldsymbol{a}_{\tau_1-1}) + \gamma\max\left\{\mathcal{M}^{\boldsymbol{\pi},\tilde{g}}v_S^{\boldsymbol{\pi},g'}(s_{\tau_1}), \max_{a_{\tau_1}\in\mathcal{A}}\left[\mathcal{R}_S(s_{\tau_k}, \boldsymbol{a}_{\tau_k}) + \gamma\sum_{s'\in\mathcal{S}}P(s'; \boldsymbol{a}_{\tau_1}, s_{\tau_1})v_S^{\boldsymbol{\pi},g}(s')\right]\right\}\right]\right]\right]$$

$$= \mathbb{E}\left[\mathcal{R}_S(s_0, \boldsymbol{a}_0) + \mathbb{E}\left[\ldots + \gamma^{l-1}\mathbb{E}\left[\mathcal{R}_S(s_{\tau_1-1}, \boldsymbol{a}_{\tau_1-1}) + \gamma\left[Tv_S^{\boldsymbol{\pi},\tilde{g}}\right](s_{\tau_1})\right]\right]\right] = v_S^{\boldsymbol{\pi},\tilde{g}}(s)$$

where the first inequality is true by assumption on $\hat{\mathcal{M}}$. This is a contradiction since $g'$ is an optimal policy for Switcher. Using analogous reasoning, we deduce the same result for $\tau'_k < \tau_k$ after which deduce the result. Moreover, by invoking the same reasoning, we can conclude that it must be the case that $(\tau_0, \tau_1, \ldots, \tau_{k-1}, \tau_k, \tau_{k+1}, \ldots,)$ are the optimal activation times.

□

## G. Proof of Theorem D.2

*Proof.* The proof of the Theorem is straightforward since by Theorem 3.4, Switcher's problem can be solved using a dynamic programming principle. The proof immediately by application of Theorem 2 in (Sootla et al., 2022).

□

