# OpenReview forum: "Learning Robust Multi-Agent Policies via Selective Adversarial Fault Induction"
_ICML.cc/2026/Conference — ICML 2026 regular_

### Official Review · Reviewer_hiec · 2026-03-02

**Soundness:** 2
**Presentation:** 3
**Significance:** 3
**Originality:** 2
**Overall Recommendation:** 3
**Confidence:** 3

**Summary:**

This paper addresses a robust multi-agent reinforcement learning framework for agent malfunction scenarios. The problem is important to the realworld robotics and the proposed method is simple and efficient. The proposed method is built on exisiting RL frameworks such as soft actor-critic. It includes a switcher that detects malfunction among agents and an adversary that executes faulty behavior. Overall, the paper is well-written and clear. The authors provide a moderate contribution in both theortetical coverage and experimental analysis.

**Compliance With Llm Reviewing Policy:**

Affirmed.

**Final Justification:**

After considering the paper and rebuttal, I am tending to keep my original scores. In my opinion, the paper is technically interesting; however, some limitations, such as neural networks theory guarantees and communication overhead latency training in real-time scenarios, still exist.

**Key Questions For Authors:**

1. What is the numerical computational and communication complexity analysis of the proposed method, especially when the number of agents increases?

2. How does the Switcher mechanism scale as the number of agents increases?

3. How do the provided failure scenarios align with the real world scenarios

**Limitations:**

The authors are encouraged to include a discussion about the limitations and future direction of their work.

**Strengths And Weaknesses:**

Strengths:

1. The paper addresses an important and timely problem in MARL.

2. The authors provide theoretical guarantee and multiple benchmarks in the deployed experiments.

3. The problem is simple and easy to integrate with existing methods. The overall organization and writing is also very good.

Weaknesses:

1. Theoretical gurantees are not clear in continous control tasks.

2. It is unclear how the Switcher mechanism scales as the number of agents increases.

3. More details on how to use and train function approximators, such as neural networks, are needed in the method formulation.

4. Complexity anaylsis is missing, which is very important in that kind of problems, especially with large number of agents.

5. Communication complexity between agents and latency analysis is also very important.

6. It is unclear how the provided failure scenarios align with the real world scenarios.

---

> ### Author Rebuttal · Authors · 2026-03-30
>
> We thank the reviewer for their feedback and for raising important points regarding theoretical scope, scalability, and practical implementation which we clarify below.
>
> **Scaling of the Switcher with number of agents**
>
> The Switcher operates over an action space of size $N+1$, corresponding to selecting one of $N$ agents or no intervention. Thus, its complexity scales linearly in the number of agents. This is significantly more efficient than methods requiring pairwise or combinatorial reasoning over agents which are known to suffer from combinatorial explosion see Gupta, Nikunj, et al. "Deep meta coordination graphs for multi-agent reinforcement learning." arXiv preprint arXiv:2502.04028 (2025).
>
> **Computational and communication complexity**
>
> MARTA introduces additional forward passes during training for the Switcher and Adversary, but reuses shared representations and replay buffers, resulting in modest overhead. Importantly, no additional communication or coordination is required at execution time, and the method preserves the decentralised execution paradigm. We will include a detailed complexity discussion.
>
> **Theoretical guarantees in continuous control**
>
> Our formal guarantees are provided under standard tabular and linear assumptions.  Extending guarantees to deep neural networks remains an open problem across RL (for example, Lim, H. "Regularized Q-learning."  NeurIPS 2024, proves convergence under linear approximation while noting the difficulty of extending results to deep networks). However, MARTA is empirically validated in continuous control settings (e.g. MPE with MADDPG) see “Comparison with robust adversarial MARL baselines” on pg 6. Extending theory to deep continuous control remains an open challenge in RL more broadly.
>
> **Function approximator details**
>
> We thank the reviewer for this suggestion. The Switcher is implemented using an actor–critic method (SAC), while the Adversary shares the backbone of the base MARL learner – this is detailed in “Details on Architecture” on pg 3. Training is performed jointly using a shared replay buffer. We will move these details into the main text for clarity.
>
> **Alignment with real-world failure scenarios**
>
> MARTA is designed to model realistic failure modes such as actuator faults, corrupted controllers, and sensor failures, which are common in robotics and distributed systems. These correspond to agent-level deviations from intended policies, rather than small perturbations. We will clarify this connection and provide additional examples.
>
> **Complexity and scalability**
>
> We thank the reviewer for emphasising the importance of complexity analysis. We have measured computational overhead across SMAC environments. MARTA introduces a modest training overhead of approximately 1.08–1.15× in wall-clock time (+8–15\%) and 1.27–1.43× in GPU memory relative to the base learner.
>
> This overhead arises from additional training components (Switcher and adversarial policies), while reusing the same backbone architectures and shared replay buffer. Importantly, MARTA does not introduce additional execution-time cost, as no online safety layer or optimisation is required at deployment. We will add these measurements and clarify the structural complexity discussion in the revision.
>
> **Practical/real world scenarios**
>
> We thank the reviewer for raising questions about practical deployment. We evaluated MARTA under varying intervention budgets in Traffic Junction. Even under strict budgets (10–25%), MARTA significantly improves performance over the baseline (e.g. **10\%: −27.9 vs −40.1 reward; 0.0108 vs 0.115 collisions**), suggesting that robustness can be achieved with infrequent interventions. Performance improves further as the budget increases, approaching full MARTA. This supports the practical feasibility of deploying MARTA under constrained resources.

---

> > ### Author Rebuttal · Reviewer_hiec · 2026-04-02
> >
> > Thanks for the authors feedback.
> >
> > For *Scaling of the Switcher with number of agents* and *Computational and communication complexity*
> >
> > Could the authors provide any theoretical or empirical evidence for these claims?
> >
> > While there is no communication overhead in execution, it is important to evaluate the training overhead as well, especially in scenarios with large number of agents.
> >
> > Thanks!

---

> > > ### Author Response · Authors · 2026-04-05
> > >
> > > We thank the reviewer for the follow-up and for emphasising the importance of empirical evidence for scalability.
> > >
> > > **Structural scaling**
> > >
> > > The Switcher introduces a control layer with action space $N+1$, corresponding to selecting one of
> > > $N$ agents or no intervention. Thus, the fault-selection decision scales linearly in the number of agents and does not require reasoning over agent pairs or subsets, while the underlying MARL backbone remains unchanged.
> > >
> > > **Empirical scaling evidence**
> > >
> > > Training time remains modest across increasing agent counts:
> > >
> > > * 3 agents (SMAC-3m): 1.10×
> > > * 8 agents (SMAC-8m): 1.15×
> > > * 10 agents (SMAC-10m vs 11m): 1.07×
> > > * 25 agents (SMAC-25m): 1.05×
> > >
> > > Across SMAC environments with 3–25 agents (e.g. 3m, 8m, 10m, 25m) and backbones (QMIX/VDN), MARTA incurs approximately 1.05–1.15× training time overhead. GPU memory usage increases with the number of agents, ranging from 1.27× to 1.61×, while remaining within a moderate range.
> > >
> > > These results provide empirical evidence that the additional cost introduced by the Switcher and adversarial components remains modest as the number of agents increases. Importantly, MARTA introduces no additional execution-time cost and preserves decentralised execution. We will include these results and clarify the complexity discussion in the revision.
> > >
> > > We would like to sincerely thank the reviewer for their comments that have improved the paper. We have incorporated these updates in the latest version.
> > >
> > > *** **UPDATE** ***
> > >
> > > Thank you again for your helpful follow-up. We have added empirical scaling results across larger agent settings (including up to 25 agents) and clarified the complexity discussion accordingly. We would be grateful if you could revisit the updated response when convenient.

---

### Official Review · Reviewer_MBmb · 2026-03-09

**Soundness:** 3
**Presentation:** 3
**Significance:** 3
**Originality:** 3
**Overall Recommendation:** 4
**Confidence:** 4

**Summary:**

The paper proposed a practical robust and fault tolerant cooperative multi agent reinforcement learning (MARL) solution. The goal to examine agent level malfunctions (e.g., actuator failures, corrupted controllers) that break the standard MARL assumption of reliable centralized training decentralized execution. The proposed MARTA system introduces two new agents (Switcher and Adversary) that controls the level of uncertainty. In each step switcher identifies the best adversarial agent to choose with certain probability (within a fixed budget) that can best hamper the overall reward, whereas the adversary agent controls and execute the behavior of the faulty agent. This N+2 player non-zero sum Markov game is played with Minimax objective to ensure that the decentralized agent policies are robust towards failures during inference time. Extensive theoretical results demonstrate the convergnece of the game under tabular and linear function approximation (albeit showing the deep neural network approximation). Empirical evaluation on Traffic Junction, Level Based Foraging, MPE SimpleTag, and SMACv2 demonstrate the supirior performance of MARTA over existing approaches and the ability to adapt in different practical implementaitons including QMIX, VDN and MADDPG. Experiments also demonstrate that MARTA is robust towards failures even when there is train-test failure distribution drift.

**Compliance With Llm Reviewing Policy:**

Affirmed.

**Key Questions For Authors:**

1. Can you summarize the computation time comparison for both train and test against the benchmark approaches?

2. Policy learning against the budget constraint is a challenging and requires larger time to converge. Have you measure the performance of such budget constraints both theoretically and experimentally?

3. Have you done any additional benchmarking of performance against deep RL methods?

**Limitations:**

Limitations and negative impact section needs to be improved. Many limitations such as different fault behavior analysis, performance analysis with deep RL are missing. Also, negative social impact in terms of adversarial traffic behavior learning could be included here.

**Strengths And Weaknesses:**

The paper addresses a practically important and challenging problem in ensuring robustness of MARL systems. While majority of the existing works studies the worst case scenario management, this paper proposed an interesting but practical setting of controlling the adversarial agents within a budget constraints using a switcher and an adversary agent, thus avoid extreme conservatism.  The key conceptual contribution is state dependent fault induction via a learned Switcher, which is fundamentally different than the existing adversarial (e.g., minimax or byzantine robust) MARL techniques. The authors provide theoretical gurantee for existence and uniqueness of a minimax value and contraction of the switching augmented Bellman operator. Moreover convergence proof is demonstrated under tabular and linear approximation setting and conditions budget constrained switching. Extensive empirical evaluation is conducted on four standard MARL benchmarks reporting superior performance on different fault regimes including (1) fixed vs. dynamic malfunctions, (2) aligned vs. shifted train–test fault distributions, and (3) random vs. worst case fault policies. Reported gains (e.g., >100% improvement in SMACv2 3m and 8m) are substantial and consistently favor MARTA over strong baselines such as QMIX, VDN, MADDPG, M3DDPG. More importantly, MARTA is designed in a plug-and-play fashion which can be on top of any MARL algorithm. While both theoretical and empirical evidences look solid, I have a few concerns regarding the paper:

1. While the theory is rigorous, it relies on standard but restrictive assumptions (bounded rewards, tabular or linear approximation). There is no formal guarantee for the deep neural network settings actually used in experiments.

2. Computation time for training and inference are not properly reported. For practical implementation, it is important to clearly report the trade-off between quality vs. computation cost. A direct computation time comparison against the benchmark approch is missing.

3. While the main theme and motivation at the early stage is about budget constraints adversarial behaviors, there were limited focus on experiments and theory on demonstrating the impact of budget constraints.

4. The Adversary is either random or worst case (softmax over −Q). More nuanced fault models (e.g., biased, delayed, partially faulty agents) are not explored. It is unclear how sensitive MARTA is to these scenarios.

5. While the design of N+2 agents Markov Game is interesting, the theoretical analysis and solution design mainly builds on standard adversarial training, so the theoretical contributions can be viewed as incremental. This point would have been mute if experimental results demonstrate superior performance of MARTA with different fault models (point 4).

---

> ### Author Rebuttal · Authors · 2026-03-30
>
> We thank the reviewer for their positive assessment of the paper and for highlighting important questions regarding computational cost, budget constraints, and robustness evaluation.
>
> **Experiments on budget constraints**
>
> We thank the reviewer for emphasising the importance of budget-constrained adversarial behaviour. We conducted additional experiments varying the switching budget in Traffic Junction. The results show a clear and consistent trade-off: even with a small budget (10–25\%), MARTA substantially improves both reward and collision rate over the baseline (e.g. **10\%: −27.9 vs −40.1 reward; 0.0108 vs 0.115 collisions**), indicating that only a small number of well-timed interventions is needed to address critical coordination failures. As the budget increases, performance improves smoothly and approaches that of full MARTA.
>
> These results support the design motivation of MARTA, demonstrating that robustness can be achieved efficiently under constrained intervention budgets. We will include this analysis in the revision.
>
> **Training/inference cost trade-off**
>
> We thank the reviewer for requesting explicit computation cost comparisons. We have measured wall-clock training time and GPU memory usage across SMAC environments. MARTA introduces a modest training overhead of approximately 1.08–1.15× in wall-clock time (+8–15\%) and 1.27–1.43× in GPU memory, depending on the environment and backbone.
>
> This reflects the additional Switcher and adversarial components, while preserving the base MARL architecture and shared replay buffer. Importantly, MARTA does not introduce additional execution-time cost, as no auxiliary safety layer or online optimisation is required at inference. We will include these results in the revised manuscript to make the trade-off explicit.
>
> **Limited fault models**
>
> MARTA is agnostic to the choice of fault policy and can naturally accommodate biased, delayed, or partial failures through the adversary policy $\sigma$ (this is explained in “Details on Architecture, pg 3). As displayed in Tabe 1, we test MARTA against a range of malfunction protocols. We will expand discussion of these extensions and include additional examples.
>
> **Deep RL theory gap**
>
> Our theoretical guarantees are established under tabular and linear approximation, which is standard in RL theory. Extending guarantees to deep neural networks remains an open problem across RL (for example, Lim, H. "Regularized Q-learning."  NeurIPS 2024, proves convergence under linear approximation while noting the difficulty of extending results to deep networks). Our results instead characterise the underlying optimisation structure that explains the empirical performance in deep MARL methods.
>
> **Theoretical contribution**
>
> We respectfully disagree with this characterisation. MARTA introduces a switching-augmented Markov game with state-dependent activation, budget constraints, and a modified Bellman operator. These structural elements are not captured by standard adversarial MARL formulations and lead to distinct equilibrium and convergence properties. We refer the reviewer to lines 173-182.

---

> > ### Author Rebuttal · Reviewer_MBmb · 2026-04-06
> >
> > I would like to thank the authors for clarifying most of my concerns. As some of my concerns like benchmarking against deep RL setup, and theoretical analysis on budget constraints settings are not fully addressed, I will keep my score as it it.

---

> > > ### Author Response · Authors · 2026-04-06
> > >
> > > We thank the reviewer for the positive assessment, thoughtful feedback and continued engagement. We wish to clarify the few remaining points.
> > >
> > > **Budget-constrained analysis**
> > >
> > > We would like to clarify the treatment of budget-constrained switching, as this appears to have been a source of confusion.
> > > The manuscript already provides a theoretical treatment of the budget-constrained setting in Section 4, where we establish convergence of the switching-augmented game under budget constraints. In response to the reviewer’s suggestion, we have now complemented this with additional empirical results that explicitly evaluate the effect of varying the intervention budget.
> > >
> > > These results show a clear and consistent trade-off: even under a 10% budget, MARTA significantly improves over the baseline (e.g. −27.9 vs −40.1 reward; 0.0108 vs 0.115 collisions), with performance approaching full MARTA as the budget increases.
> > >
> > > Together, these results provide both theoretical justification and empirical validation of the budget-constrained formulation. We will revise the manuscript to make this connection more explicit. We would like to thank the reviewer for the suggested empirical evaluation which has further strengthened the paper.
> > >
> > > **Deep RL benchmarking**
> > >
> > > We wholeheartedly agree that evaluation with deep RL methods is important. We would like to clarify that MARTA is already evaluated with deep MARL approaches in the current manuscript, including MADDPG (actor-critic) and EIR (Byzantine-robust MARL) (see Section 5).
> > >
> > > We will revise the manuscript to make this more explicit, as this may not have been sufficiently emphasised in the current presentation.
> > >
> > > *** **UPDATE** ***
> > >
> > > Thank you again for your constructive feedback. We have provided additional clarification on both the budget-constrained setting (including new experiments) and the deep RL benchmarking (MADDPG/EIR) in our latest response. We would appreciate it if you had an opportunity to review these updates.

---

### Official Review · Reviewer_ZMG1 · 2026-03-10

**Soundness:** 3
**Presentation:** 2
**Significance:** 2
**Originality:** 3
**Overall Recommendation:** 4
**Confidence:** 2

**Summary:**

The paper proposes a method to enhance the robustness of multi-agent reinforcement learning systems by inducing agent malfunctions during training.

**Compliance With Llm Reviewing Policy:**

Affirmed.

**Key Questions For Authors:**

1) Can the approach work for policy-based methods?
2) What is the computational burden of the proposed method?

**Limitations:**

1) The method is specific to value-based methods, while most RL systems are now policy-based.
2) The scale at which experiments are conducted is not very clear to me, making the generalizability of the method unclear.

**Strengths And Weaknesses:**

Strengths:

The paper provides theoretical guarantees for the tabular and linear function approximation cases, which are the cases where theoretical guarantees can generally be established in RL. This is achieved with a nice Bellman optimality convergence condition, with a min/max modification. The paper also tests and validates the performance of the proposed method on multiple diverse benchmarks, although multiple are small in scale.


Weaknesses:

The method is restricted to the value-based regime. The update is based on a max/min over the actions, which might not be feasible if the action space is continuous or extremely large. This also raises an important concern about the computational feasibility of this method, especially in deployment scenarios.

The choice of baselines is questionable. Constrained RL contains prominent baselines like [1], which can be integrated into the policy-based algorithms. You mention that [1] has convergence and stability issues, but can this not be demonstrated numerically on the experiments?


[1] Multi-Agent Constrained Policy Optimisation, Gu et Al.

---

> ### Author Rebuttal · Authors · 2026-03-30
>
> We thank the reviewer for their comments, especially regarding applicability beyond value-based methods, computational feasibility, and baseline comparisons.
>
> **Scale of experiments**
>
> We clarify that MARTA is evaluated not only on small-scale benchmarks but also on SMACv2, which introduces larger agent populations, partial observability, and non-trivial coordination requirements (see Sec 5.). This is a standard benchmark within MARL papers. The observed gains (e.g. >100% in SMAC-3m) indicate that the method scales beyond small environments (see Figs 2, 4 and Table 3).
>
> **Feasibility/Practicability**
>
> We thank the reviewer for raising the question of budget-constrained performance. We performed additional experiments varying the switching budget in Traffic Junction. Even with a limited budget (10–25\%), MARTA significantly outperforms the baseline (e.g. **10\%: −27.9 vs −40.1 reward; 0.0108 vs 0.115 collisions**), demonstrating that effective robustness can be achieved without frequent interventions. Increasing the budget yields further improvements, approaching full MARTA performance. This suggests that MARTA can operate effectively under realistic resource constraints.
>
> **Computational feasibility**
>
> We thank the reviewer for raising the question of computational feasibility. We have measured wall-clock training time and GPU memory usage across SMAC environments. MARTA incurs a modest overhead of approximately 1.08–1.15× in training time (+8–15\%) and 1.27–1.43× in GPU memory relative to the base learner.
>
> Structurally, MARTA preserves the underlying MARL training pipeline and reuses shared replay buffers, avoiding duplicated environment interaction. The additional cost arises from training the Switcher and adversarial policies. Importantly, no additional computation is introduced at execution time, as MARTA does not require online optimisation or safety filtering at deployment.
>
> **Restricted to value-based methods**
>
> We respectfully clarify that MARTA is not restricted to value-based methods. While our theoretical analysis is presented in a value-based framework, MARTA is implemented and evaluated with both value-based (QMIX, VDN) and actor–critic methods (MADDPG) – see “Comparison with robust adversarial MARL baselines” on pg 6. In addition, the Switcher itself is trained using a policy-based method (SAC). This demonstrates that MARTA is compatible with modern policy-based MARL approaches.
>
> **Continuous action / max-min concern**
>
> We agree that explicit max and min over large or continuous action spaces is generally not tractable. In practice, MARTA does not rely on enumeration. The adversarial component is implemented as a learned policy $\sigma$, so the “min” so is realised through parametric policy optimisation rather than explicit search. This is standard in adversarial RL e.g., Robust Adversarial Reinforcement Learning (Pinto et al., ICML 2017) formulates training as a two-player minimax game in which both agent and adversary are trained jointly, rather than solving the inner minimisation analytically. We will clarify this distinction between theoretical formulation and practical implementation.
>
> **Constrained RL Baselines**
>
> Constrained MARL methods address a formulation where safety is enforced through explicit constraints on policies or trajectories, typically via constrained optimisation e.g. *Multi-Agent First Order Constrained Optimization in Policy Space* (NeurIPS 2023). In contrast, MARTA targets a distinct setting of endogenous, state-dependent agent failures, learning where malfunctions most impact coordination rather than constraining behaviour a priori. We agree that clarifying this distinction will improve positioning and will expand the discussion accordingly.

---

> > ### Author Rebuttal · Reviewer_ZMG1 · 2026-03-31
> >
> > My concerns have been addressed. I thank the authors for the detailed rebuttal. As a consequence, I will increase my score, but I will need to reiterate my limited expertise in multi-agent reinforcement learning.

---

### Official Review · Reviewer_DrFy · 2026-03-13

**Soundness:** 3
**Presentation:** 1
**Significance:** 4
**Originality:** 3
**Overall Recommendation:** 4
**Confidence:** 3

**Summary:**

This paper investigates learning multi-agent policies that are robust to faults and failures of individual agents. To address this problem, the authors propose a framework that introduces two additional agents, a switcher and an adversary, which are trained jointly with the original agents to simulate adversarial faults. The switcher agent learns to select agents on which faults should be imposed, while the adversary agent induces faulty behavior in the selected agent with the objective of minimizing the discounted reward while accounting for a cost associated with inducing faults. In contrast, the cooperative agents aim to maximize the discounted reward. The proposed mechanism can be integrated with existing state-of-the-art multi-agent reinforcement learning (MARL) algorithms as an extension. The authors further show that the switcher–adversary mechanism induces a non-zero-sum Markovian game and provide theoretical guarantees, including convergence to a unique equilibrium. To support the theoretical analysis, the authors conduct experiments across several benchmark environments. The results indicate improved performance of the proposed framework (MARTA) compared to existing approaches.

**Compliance With Llm Reviewing Policy:**

Affirmed.

**Final Justification:**

I thank the authors for their effort. The main technical questions are more or less addressed through the authors' rebuttals. While it seems to be that authors are willing to improve the presentation, it would have been beneficial if authors had shared some evidence on such revisions, especially, based on my original opinion on the presentation.
As such, I would like to keep my favorable score as it was.

**Key Questions For Authors:**

1.	The experimental configuration states that, at each timestep, an agent malfunctions with probability $p$ and executes a corrupted policy $σ_i$. This appears to conflict with the proposed switcher–adversary mechanism, which is designed to learn when and where to induce faults. Could the authors clarify how these two mechanisms interact? For example, is the switcher responsible for selecting the malfunctioning agent while $p$ controls the activation probability, or are malfunctions generated independently of the learned switcher policy?
2.	The paper claims that the switcher–adversary mechanism introduces only marginal computational overhead and can be integrated with existing MARL algorithms. Could the authors provide more concrete details, such as additional training time, memory overhead, or parameter counts, compared to the base MARL algorithms, especially when the system scales?
3.	Only the “Hard difficulty” setting suggests that the malfunction distribution during testing differs from that used during training. In practice, malfunctions are often uncontrolled and unpredictable at deployment time. Could the authors elaborate on how the framework performs when the malfunction distribution during testing deviates significantly from the one encountered during training?
4.	Additionally, could the authors revise the flow of presentation so that readers may able to understand the concept, notations, and ideas in a smooth flow of reading?

**Limitations:**

Yes

**Strengths And Weaknesses:**

**Soundness:**

The paper provides a clear formalization of agent malfunction robustness in MARL through a game-theoretic framework. The authors present theoretical guarantees, including contraction of the Bellman operator, existence and uniqueness of the minimax value, and convergence to a Markov perfect equilibrium. Empirical evaluation is conducted across multiple benchmark environments, including cooperative and coordination-intensive tasks, which helps demonstrate the generality of the proposed approach. However, several concerns remain:
- According to Fig. 1, the switcher–adversary mechanism appears to intervene mostly at the entry and exit points of the map rather than at the junction, where collisions and coordination failures are more likely. To reduce the reward effectively, would it not be more beneficial for the adversarial mechanism to intervene at such critical points?
- In Footnote 3, the equilibrium definition would be more precise if phrased as: “... no agent can improve the expected return by **unilaterally** changing its policy ...”.
- Several notations are undefined, including $L_2$, $\Psi$, dimension $p$  in Theorem 3.4, as well as support spaces $\mathbb{L}$, $\mathbb{A}$, $\mathbb{B}$ in Lemma E.5.
- The statement in Experiment Configurations that “At each timestep $t$, with probability $p$, one agent is selected to malfunction and execute a corrupted policy $σ_i$ at $t$” appears to contradict the role of the switcher–adversary mechanism, which is supposed to learn when and where to induce faults.
- Only the “Hard difficulty” setting suggests that training and testing use different malfunction distributions. However, in realistic deployments, malfunctions are not controlled during testing. Could the authors clarify the rationale behind this design?
- Lemma E.3 contains a typographical error: the function definition$g:\mathcal{y} \to \mathbb{R}$ appears to be missing $g$.
- After Lemma E.9, it is unclear why $\mathcal{P}_{ss’}^a$ depends on $s’$, given that the right-hand side includes a summation over all possible $s’$.
- On page 21, the first max⁡(⋅) term contains a typographical error where $pi$ should be replaced by $\boldsymbol{a}$.
- On the same page, in the proof of (ii), the operator $\hat{\mathcal{M}}$ is defined in terms of $Q$, whereas the derivation uses $\mathcal{P}_{ss’}^a v_S$. Should this instead follow from taking the expectation over the adversary policy $sigma$ rather than directly applying the transition operator?

**Presentation:**

The paper contains several sections that are difficult to follow due to the flow and repeated explanations.
- Figure 1 lacks a sufficiently descriptive caption.
- Page 3, column 1, lines 128–132 introduce several policies in a way that makes it difficult to determine which policies correspond to which agents. Rephrasing would improve clarity.
- Page 3, column 1, lines 155–157 repeat ideas that were already introduced earlier.
- Page 3, column 2, lines 120–121 contain an unclear sentence: “… nor a team game can occasion convergence issues (Shoham & Leyton-Brown, 2008).”
- The manuscript frequently references future results, claims, or theorems, which makes the reading flow difficult. In particular, proofs and related lemmas are distributed across the main text and appendices in a way that complicates the logical progression.
- Several notation definitions appear only in the appendix, even though the notation is used in the main text. This requires readers to frequently move between sections.
- Under Environments (Point 2), the phrase “… performing the ‘collectqaction is greater …’” appears to be incomplete or unclear.
- Maintain consistent terminology and notation, e.g., using either Fig. or Figure, referencing equations as (1) rather than 1, using consistent quotation marks in LaTeX, and avoiding appendix labels such as A.x for definitions.
- Avoid repeating figure captions as figure titles.

**Significance:**

Robustness to malfunctioning agents is an important problem for real-world MARL deployments. By focusing on state-dependent and selective fault induction, the paper highlights a potential weakness of existing MARL training paradigms. However, although the computational overhead of the proposed method is claimed to be marginal, the paper does not provide quantitative or qualitative analysis of the additional computational cost.

**Originality:**

Modeling robustness through a fault-inducing switcher–adversary mechanism is conceptually novel in the MARL literature. The integration of fault injection, adversarial control, and MARL training represents a creative combination of ideas.

---

> ### Author Rebuttal · Authors · 2026-03-30
>
> We thank the reviewer for their thoughtful and detailed feedback, particularly regarding clarity of presentation, experimental design, and the role of the switcher–adversary mechanism. We have responded to these comments below.
>
> **Computational overhead / significance**
>
> We thank the reviewer for highlighting the need to quantify computational overhead. We have measured wall-clock training time and GPU memory usage across SMAC environments. MARTA introduces a modest training overhead of approximately +8–15\% wall-clock time ($\approx$ 1.08–1.15×) relative to the base learner, with GPU memory usage increasing by approximately 1.27-1.43×, depending on the environment and backbone (QMIX/VDN).
>
> This overhead arises from training the Switcher and adversarial components, while reusing the same backbone architectures and shared replay buffer (as described in the method section). Importantly, MARTA does not introduce additional execution-time cost, as no safety filter or online optimisation is required at deployment. We will include these measurements in the revised manuscript.
>
> **Clarification of malfunction probability $p$ vs Switcher**
>
> We thank the reviewer for the observation and clarify that $p$ and the Switcher operate in different phases. While training MARTA, malfunctions are entirely determined by the learned Switcher policy $g(s)$ which selects whether and which agent to corrupt; no exogenous stochastic triggering is used. \emph{During evaluation} (of all methods), we introduce an external malfunction process: with probability $p$, an agent executes a corrupted policy $\sigma$, defining controlled test-time fault regimes (e.g. aligned vs shifted distributions). Table 1 gives the full set of evaluation malfunction configurations. We will revise the manuscript to make this distinction explicit.
>
> **Switcher interventions**
>
> We thank the reviewer for this observation. The Switcher does not exclusively target entry/exit regions; rather, it concentrates interventions at coordination-critical states, including both intersections and entry/exit points (as shown in Fig. 1b). While junctions are indeed collision-prone, failures at entry/exit points can propagate downstream and disrupt coordination over longer horizons. The Switcher therefore learns to target states that maximise expected degradation in future coordination, not only immediate collisions. We will clarify this distinction in the revision.
>
> **Train-test mismatch**
>
> The “Hard” setting is not meant to model controlled faults, but to evaluate distributional robustness under adversarial shift, which is critical in real deployments. This setting therefore tests whether policies generalise to unseen fault distributions, which is a standard robustness criterion. This aligns with prior work on robustness in RL that evaluates performance under distribution shift or out-of-distribution conditions, rather than matched train–test settings (e.g. domain randomisation and generalisation studies such as (Packer et al., 2018).
>
> **Notation and typography**
>
> We thank the reviewer for carefully identifying these issues. We will correct all typographical errors and ensure that all notation used in the main text is defined locally to improve readability. In particular,  we will (i) reduce forward references, (ii) introduce key notation earlier in the main text, and (iii) improve figure captions and explanations to ensure a smoother reading flow.

---

> > ### Author Rebuttal · Reviewer_DrFy · 2026-04-03
> >
> > Thank you for the responses. I have few more concerns as follows:
> >
> > **On switcher interventions:**
> >
> > As a matter of fact, Figure 1 is not described nor referred to in the original body of the text. As a reader, how I interpreted Figure 1 is that four agents entering to the junction from four direction using right lanes, moving forward, yet unclear whether they will simply move forward towards the exit or take turns. Under which, it is unclear why entering and exiting points have significant impact on the performance compared to the points that are close to the intersection. In my opinion, when agents are about to collide, failures are highly significant compared to the failures at the entry points where they can adjust. Moreover, failures at the exits should have the least significant impact, which contradicts Fig. 1(b) that shows the highest intervention occurring at the exist points. Maybe authors require some clear details with substantial revisions as this is the earliest point where the idea is conveyed, in which, plays a critical role.
> >
> > **Computational overhead / significance:**
> >
> > While results on training time and computation are briefed, how they scale as the system scales remains unaddressed.
> >
> > **Train-test mismatch**
> >
> > In the manuscript, authors claim that “Easy (Level 1) settings correspond to aligned train–test distributions where a single fixed agent may malfunction with probability p” and “Hard (Level 3) settings involve train–test shifts, either in the fault distribution, malfunction probability, or fault policy”. These claims somewhat disagree with the justification. Certain rephrasing may require to avoid the mismatch.
> >
> > **Presentation:**
> >
> > It is bit unclear how the manuscript is thoroughly revised to convey the logical flow of the paper.
> >
> > ***
> > As a matter of fact, I have no sufficient evidence to increase the score.

---

> > > ### Author Response · Authors · 2026-04-05
> > >
> > > We thank the reviewer for the detailed follow-up and clarify the remaining points below.
> > >
> > > **Switcher interventions (Fig. 1)**
> > >
> > > Switcher interventions (Fig. 1). We agree that Fig. 1 was insufficiently described and will revise both the caption and main text accordingly. The key point is that the Switcher does not optimise for immediate collisions, but for long-horizon degradation of coordination.
> > >
> > > In Traffic Junction, disruptions at entry points can alter traffic flow and timing, propagating downstream and increasing congestion and collision risk later in the episode. Similarly, failures near exits can delay lane clearance and affect trailing agents. Thus, the Switcher learns to intervene at states with high downstream impact, rather than only at immediate collision points. We will clarify this temporal effect and explicitly reference Fig. 1 in the revised manuscript.
> > >
> > > **Computational overhead and scaling**
> > >
> > > We agree that scaling behaviour should be demonstrated empirically and thank the reviewer for this suggestion. In addition to earlier results (3m and 8m), we evaluated larger agent settings. The following values report the training-time multiplier of MARTA relative to the base learner:
> > >
> > > * 3 agents (SMAC-3m): 1.10×
> > > * 8 agents (SMAC-8m): 1.15×
> > > * 10 agents (SMAC-10m vs 11m): 1.07×
> > > * 25 agents (SMAC-25m): 1.05×
> > >
> > > This trend is consistent - across SMAC environments with 3–25 agents (e.g. 3m, 8m, 10m, 25m) and backbones (QMIX/VDN), MARTA incurs approximately 1.05–1.15× training time overhead. GPU memory usage increases with the number of agents, ranging from 1.27× to 1.61×, while remaining within a moderate range.
> > >
> > > **Train-test mismatch**
> > >
> > > We agree the phrasing may be unclear. During training, malfunctions are fully determined by the Switcher policy. The “Easy/Hard” distinction refers only to evaluation settings: Easy corresponds to aligned fault distributions, while Hard introduces shifts in fault probability, agent selection, or fault policy. We will revise the wording to clearly separate training and evaluation.
> > >
> > > **Presentation**
> > >
> > > We acknowledge the concerns regarding clarity and will revise the manuscript to improve logical flow, reduce forward references, and introduce key concepts earlier, including clearer integration of Fig. 1.
> > >
> > > We would like to sincerely thank the reviewer for their comments that have improved the paper. We have incorporated these updates in the latest version.
> > >
> > > *** **UPDATE** ***
> > >
> > > Thank you again for your detailed follow-up. We have addressed the points regarding Fig. 1, scaling behaviour (including additional larger-scale experiments), and the train–test clarification in our latest response. We would greatly appreciate it if you had a chance to take another look when convenient.

---

### Decision · Program_Chairs · 2026-04-30

**Decision:**

Accept (regular)

**Comment:**

This paper investigates MARL learning multi-agent policies that are robust to agent faults by introducing two additional agents, a switcher and an adversary, which are trained jointly with the original agents to induce faults during training. The reviewers appreciated the novelty and combination of ideas in this approach, including the budget constraints that help to avoid being overly conservative.

The reviewers noted the development of theoretical guarantees for the cases of MARL with tabular and linear functions (as is common in RL literature), and there remains a theory-practice gap. Also noted, there are no theory guarantees for neural network based implementation and continuous action spaces (which may be a larger question, but worthy of discussion at least).

The reviewers appreciate the additional results shown in rebuttal discussions, but recommend a more detailed complexity and scalability analysis generally, specifically to include computation and communications during training.

During rebuttal discussions, the authors have clarified and noted the need for revisions, however the reviewers remain concerned that the original submitted paper has significant issues with logical flow, presentation and clarity.  They conclude that the paper should be thoroughly and carefully revised going forward.